# Nucleus-translocated GCLM promotes chemoresistance in colorectal cancer through a moonlighting function

Jin-Fei Lin[1,2,8], Ze-Xian Liu [1,8], Dong-Liang Chen [1,8], Ren-Ze Huang[1,8], Fen Cao[3], Kai Yu[4], Ting Li[5], Hai-Yu Mo[1], Hui Sheng[1], Zhi-Bing Liang[6], Kun Liao[1], Yi Han[1], Shan-Shan Li[6], Zhao-Lei Zeng [1], Song Gao [1], Huai-Qiang Ju [1,6] & Rui-Hua Xu [1,7]

Metabolic enzymes perform moonlighting functions during tumor progression, including the modulation of chemoresistance. However, the underlying mechanisms of these functions remain elusive. Here, utilizing a metabolic clustered regularly interspaced short palindromic repeats (CRISPR)-Cas9 knockout library screen, we observe that the loss of glutamate-cysteine ligase modifier subunit (GCLM), a rate-limiting enzyme in glutathione biosynthesis, noticeably increases the sensitivity of colorectal cancer (CRC) cells to platinum-based chemotherapy. Mechanistically, we unveil a noncanonical mechanism through which nuclear GCLM competitively interacts with NF-kappa-B (NF-κB)-repressing factor (NKRF), to promote NF-κB activity and facilitate chemoresistance. In response to platinum drug treatment, GCLM is phosphorylated by P38 MAPK at T17, resulting in its recognition by importin a5 and subsequent nuclear translocation. Furthermore, elevated expression of nuclear GCLM and phospho-GCLM correlate with an unfavorable prognosis and poor benefit from standard chemotherapy. Overall, our work highlights the essential nonmetabolic role and posttranslational regulatory mechanism of GCLM in enhancing NF-κB activity and subsequent chemoresistance.

Oxaliplatin-based chemotherapy remains the main first-line chemotherapeutic regimen for patients with unresectable colorectal cancer (CRC)[1], although immunotherapy is currently the most promising strategy for successfully treating tumors[2]. However, the associated side effects and chemoresistance are the main factors contributing to treatment failure, and effective predictive markers to identify patients who would benefit from such treatments are lacking. Because of these limitations, patients can experience tumor recurrence and metastasis[3]. Therefore, there is an urgent need to elucidate the mechanisms of oxaliplatin-based chemoresistance and develop effective strategies to enhance chemosensitivity to improve the prognosis and survival of patients with CRC.

[1]Department of Medical Oncology, State Key Laboratory of Oncology in South China, Guangdong Provincial Clinical Research Center for Cancer, Sun Yat-sen University Cancer Center, Guangzhou, PR China. [2]Department of Clinical Laboratory, Sun Yat-Sen University Cancer Center, Guangzhou, PR China. [3]Department of Biochemistry, Zhongshan School of Medicine, Sun Yat-sen University, Guangzhou, PR China. [4]Department of Genomic Medicine, The University of Texas MD Anderson Cancer Center, Houston, TX, USA. [5]Department of Gastroenterology and Urology, Hunan Cancer Hospital/The Affiliated Cancer Hospital of Xiangya School of Medicine, Central South University, Changsha, PR China. [6]Department of Clinical Oncology, Shenzhen Key Laboratory for Cancer Metastasis and Personalized Therapy, The University of Hong Kong-Shenzhen Hospital, Shenzhen, PR China. [7]Research Unit of Precision Diagnosis and Treatment for Gastrointestinal Cancer, Chinese Academy of Medical Sciences, Guangzhou, PR China. [8]These authors contributed equally: Jin-Fei Lin, Ze-Xian Liu, Dong-Liang Chen, Ren-Ze Huang. ✉e-mail: juhq@sysucc.org.cn; xurh@sysucc.org.cn

Metabolic reprogramming is a crucial hallmark of malignant cells. Cancer cells must reprogram their metabolic and energy production networks to meet the requirements for exponential proliferation and maintain critical cellular processes[4,5]. Previous studies, including ours[3,6–9], have indicated that metabolic enzymes modulated by critical signaling pathways play indispensable roles in metabolic reprogramming, and that the aberrant expression or activity of these enzymes is closely related to tumor progression and chemoresistance. Given the essential functions of metabolic enzymes in normal cells, targeting them directly for cancer treatment has inherent limitations. However, accumulating evidence has demonstrated that multiple metabolic enzymes can also support malignant transformation via noncanonical (moonlighting) functions in addition to their canonical roles[10–12]. Hence, targeting these moonlighting functions of metabolic enzymes remains an attractive approach for the diagnosis, monitoring, and treatment of tumors.

The metabolic enzyme glutamate-cysteine ligase modifier subunit (GCLM) is the rate-limiting enzyme in the biosynthesis of glutathione (GSH), which contributes to many pathological conditions. As a modifier subunit, GCLM directly interacts with the glutamate cysteine ligase (GCL) catalytic subunit (GCLC) to modify the catalytic efficiency of GCLC, increasing GSH synthesis to maintain a favorable redox balance[13]. A previous study indicated that GCLM is closely associated with the drug resistance of tumor cells, and studies on GCLM have focused primarily on the function of this metabolic enzyme in the cytoplasm[14,15]. However, its nonmetabolic functions and underlying mechanism in chemoresistance and cancer progression remain largely unexplored.

In this work, we reveal a nonmetabolic mechanism through which GCLM directly regulates nuclear factor kappa-B (NF-κB) activity, by interacting with NF-κB-repressing factor (NKRF), to endow CRC cells with resistance to platinum-based chemotherapy. P38 MAPK is the upstream mediator that phosphorylates GCLM at T17 following platinum drug treatment; this phosphorylation event is required to increase the interaction of GCLM with importin a5 and its nuclear accumulation. Furthermore, nuclear GCLM and phospho-GCLM at T17 are prognostic indicators of an unfavorable prognosis and poor benefit from standard chemotherapy in CRC patients. Our findings strongly highlight the moonlighting function of GCLM and suggest that a rational combination of P38 MAPK–nuclear GCLM–NF-κB axis inhibition and platinum-based chemotherapy is an efficient approach for the treatment of CRC.

## Results

### GCLM depletion enhances the chemosensitivity of CRC cells to oxaliplatin

To identify the crucial molecules affecting the sensitivity of CRC cells to chemotherapy, we performed a loss-of-function screen with a metabolic enzyme-encoding gene-based clustered regularly interspaced short palindromic repeats (CRISPR)-Cas9 library in the presence of oxaliplatin or PBS (Fig. 1a). A total of 52 essential metabolic enzymes were identified by comparing the top 100 genes in the three oxaliplatin-treated groups with those in the control group (Fig. 1b and Supplementary Data 1). We conducted a cell viability assay to confirm the role of the 10 genes in oxaliplatin sensitivity among the 52 candidate genes, that were highly expressed in patients with colon and rectum adenocarcinoma compared with normal patients according to the Gene Expression Profiling Interactive Analysis (GEPIA) database. The results showed that knocking out GCLM had the greatest effect on increasing the sensitivity of HCT116 cells to oxaliplatin (Fig. 1c and Supplementary Fig. 1a). In addition, compared with inhibiting GCLC, inhibiting GCLM was more effective at reducing cell viability following oxaliplatin treatment (Supplementary Fig. 1b, c). Next, we found that GCLM depletion significantly decreased the $IC_{50}$ values of oxaliplatin and increased the apoptosis in HCT116 and DLD1 cells treated with different concentrations of oxaliplatin (Fig. 1d, e and Supplementary Fig. 1d). These results indicate that GCLM may be a key molecule contributing to chemoresistance in CRC.

We used subcutaneous cell-derived xenograft (CDX) models in vivo to further confirm the role of GCLM in the chemosensitivity of CRC. Consistent with the in vitro results, the growth rates and weights of the xenograft tumors, formed from implanted GCLM-knockdown HCT116 and DLD1 cells, were reduced compared with those of tumors formed from the corresponding control cells (Fig. 1f and Supplementary Fig. 1e, f). Compared with either of the monotherapies, the combination of GCLM inhibition and oxaliplatin treatment achieved the best suppressive effects on the CDX models (Fig. 1f and Supplementary Fig. 1e, f). To better emulate the physical tumor microenvironment, we established two patient-derived xenograft (PDX) tumor models as previously reported (Supplementary Fig. 1g)[16]. Consistently, we observed decreased tumor volumes and weights after GCLM was silenced through intratumoral RNA interference (RNAi) injection, and the combination of GCLM inhibition and oxaliplatin treatment group resulted in the greatest reductions in tumor volume and weight in both PDX #1 and PDX #2 models (Fig. 1g, h and Supplementary Fig. 1h, i). In addition, a pathological analysis with hematoxylin-eosin (H&E) staining revealed increased necrosis in the GCLM-depleted group, oxaliplatin-treated group, and the combination treatment group, which indicated a robust therapeutic response (Fig. 1i and Supplementary Fig. 1j). Similarly, the combination treatment group also showed presented decreased cell proliferation (Ki67+ staining), increased cell death (Tunel+ assay) and more loose and regular collagen fibers (blue in Masson's trichrome staining)[17], which surpassed the efficacy of GCLM depletion or oxaliplatin treatment alone (Fig. 1i, j and Supplementary Fig. 1j, k). Taken together, the results suggest that GCLM inhibition notably enhances the chemosensitivity of CRC cells to oxaliplatin.

### GCLM translocates to the nucleus up platinum drug treatment

We then evaluated the clinical relevance of GCLM in CRC patients from Sun Yat-sen University Cancer Center (SYSUCC). The results showed that GCLM expression was notably increased in CRC tissues compared with adjacent normal tissues (Fig. 2a). Moreover, we categorized the GCLM level as low or high compared with the median value and found that high GCLM expression was associated with shorter overall survival and disease-free survival times (Fig. 2b). In addition, we validated the clinical correlation between GCLM expression and the response of patients with CRC to oxaliplatin-based chemotherapy (FOLFOX or XELOX regimen)[18], and found that GCLM staining was increased in patients who exhibited lesser benefit from standard chemotherapy (Fig. 2c). Unexpectedly, IHC staining revealed that GCLM, which is a cytoplasmic metabolic enzyme, was diffusely localized in the nucleus in tissues, especially from patients with lesser benefit from chemotherapy (Fig. 2c). Because GCLM directly interacts with GCLC to modify the catalytic efficiency of GCLC in the cytoplasm[13], we inhibited its enzymatic activity using the GCLC inhibitor buthionine-(S,R)-sulfoximine (BSO) or mutating cysteine 193/194 of GCLM to alanine (C193/194 A), which disrupts disulfide bonds between GCLM and GCLC[13,19], to explore whether GCLM affects chemosensitivity via its metabolic enzyme activity. Compared with the re-expression of shRNA-resistant (r) wild-type (WT) GCLM in GCLM-knockdown cells, the re-expression of C193/194 A mutant rGCLM or BSO treatment indeed failed to reverse the reduced GSH level induced by GCLM knockdown (Supplementary Fig. 2a, b). However, Compared with the control group, BSO treatment or re-expression of C193/194 A mutant rGCLM still increased cell viability as well as rGCLM WT in both the presence and absence of oxaliplatin treatment (Fig. 2d and Supplementary Fig. 2c). In addition, oxaliplatin and other commonly utilized platinum drugs notably promoted the accumulation of GCLM in the nucleus in HCT116 and DLD1 cells, with a slight change in the total

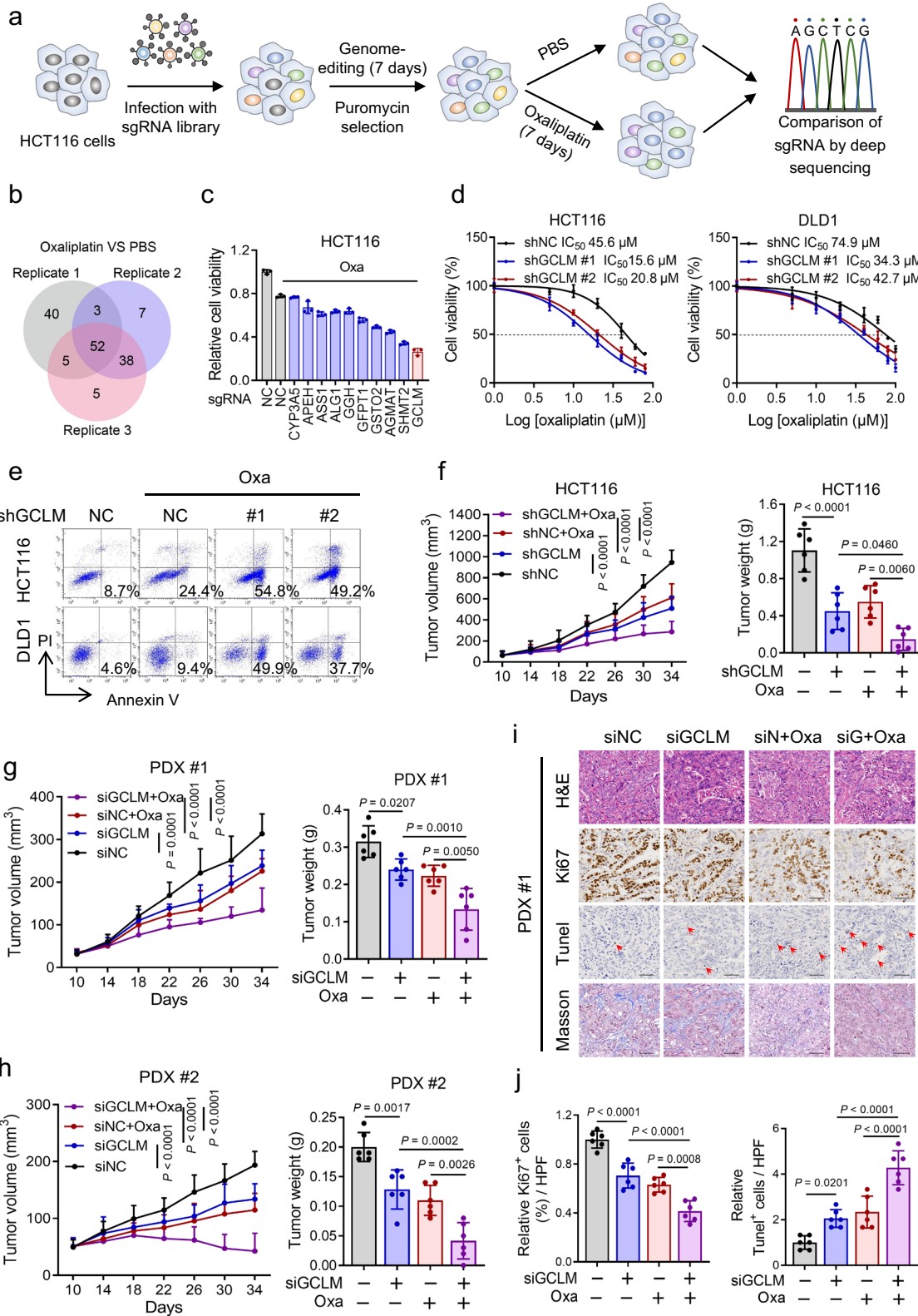

GCLM protein level (Fig. 2e, f and Supplementary Fig. 2d, e). These results reveal that the expression of GCLM might be a reliable indicator for predicting the chemosensitivity of CRC patients and that GCLM may have a moonlighting function, based on its noncanonical localization.

Previous studies have focused on the function of GCLM in the cytoplasm[13]. To test whether nuclear GCLM contributes to resistance to platinum-based chemotherapy in CRC cells, we selectively increased the GCLM level in the nucleus by expressing rGCLM or the C193/194 A mutant fused to a nuclear localization signal (rGCLM-NLS) in endogenous GCLM-knockdown HCT116 cells, leaving the level of cytoplasmic GCLM unchanged (Supplementary Fig. 2f). The results showed that overexpressing rGCLM-NLS had a negligible effect on the level of GSH (Supplementary Fig. 2g). However, overexpression of nuclear GCLM could increase the cell viability and decrease the apoptosis rate, as well as in the cells treated with oxaliplatin or cisplatin (Fig. 2g, h and

**Fig. 1 | GCLM depletion enhances the chemosensitivity of CRC cells to oxaliplatin. a** Diagram showing the strategy for the CRISPR-Cas9 screen in HCT116 cells under the treatment with PBS or oxaliplatin (10 μM, 7 days). The genomic DNA of control and treated cells was isolated and amplified for comparison of sgRNAs via deep sequencing. **b** Venn diagram showing the top 100 downregulated genes and the overlapping genes in the three oxaliplatin-treated groups compared with the control group (Supplementary Data 1). **c** Cell viability of HCT116 cells with or without oxaliplatin treatment (40 μM, 24 h) after each candidate gene (CYP3A5, APEH, ASS1, ALG1, GGH, GFPT1, GSTO2, AGMAT, SHMT2 or GCLM) was knocked out. **d** $IC_{50}$ Analysis of oxaliplatin in HCT116 and DLD1 cells treated with different concentrations of oxaliplatin for 48 h. **e** Annexin V/PI staining analysis was used to evaluate the percentages of apoptotic cells among control and GCLM-knockdown CRC cells treated with PBS or oxaliplatin (40 μM for HCT116 cells and 80 μM for DLD1 cells, 24 h). **f** Statistical analysis of CDX tumor volumes and weights in nude mice after the implantation of GCLM-knockdown or control HCT116 cells ($2 \times 10^6$),

followed by intraperitoneal injections of PBS or oxaliplatin (5 mg/kg). Statistical analysis of the tumor volumes and weights in nude mice in the PDX #1 (**g**) and PDX #2 (**h**) models, followed by intratumoral injections of in vivo-optimized GCLM inhibitor (siGCLM) or the control siRNA (5 nmol per injection), and intraperitoneal injections of PBS or oxaliplatin (5 mg/kg). **i** Representative images of H&E, IHC staining for Ki67, Tunel staining and Masson's trichrome staining in PDX #1-based paraffin-embedded subcutaneous tumor sections. The red arrowheads indicate the positive cell of Tunel staining. Scale bar = 50 μm. **j** Quantification of the proliferation index (Ki67 staining) and apoptotic index (Tunel assay) in the PDX #1 models. n = 3 biologically independent experiments in (**c, d**) and n = 6 mice in (**f–h, j**). All the data are presented as the mean ± S.D. The *P* values were calculated by one-way ANOVA (**f–h** right and **j**), and two-way ANOVA (**f–h** left). Oxa oxaliplatin, CDX cell-derived xenograft, PDX patient-derived xenograft, CRC colorectal cancer, ANOVA analysis of variance.

---

Supplementary Fig. 2h, i). Moreover, the C193/194 A mutation had little effect on the function of nuclear GCLM (Fig. 2g and Supplementary Fig. 2h). Consistently, nuclear GCLM overexpression increased the xenograft tumor growth rate and weight in vivo (Fig. 2i). Under the treatment of oxaliplatin, nuclear GCLM overexpression observably weakened the therapeutic efficiency of oxaliplatin compared with the control group (Fig. 2i). Overall, our results show that platinum drug treatment promotes the nuclear accumulation of GCLM, which performs a moonlighting function in decreasing the sensitivity of CRC cells to platinum-based chemotherapy independent of its metabolic enzyme activity.

## Nuclear GCLM interacts with NKRF to orchestrate NF-κB activity and chemoresistance

To further explore the potential function of nuclear GCLM, Flag-GCLM was expressed in HEK293T cells to increase its expression, and the cytoplasmic and nuclear fractions were used for immunoprecipitation (IP) after treatment with oxaliplatin. The abundances of GCLM-associated proteins in 55–130 kDa range were higher in the nuclear lysate than in the cytoplasmic lysate (Fig. 3a). Liquid chromatography-tandem mass spectrometry (LC-MS/MS) analysis of the nuclear GCLM-associated proteins and endogenous Co-IP identified NKRF as the candidate target of nuclear GCLM (Fig. 3b and Supplementary Fig. 3a, b). Additionally, the direct interaction between NKRF and GCLM was verified by a glutathione S-transferase (GST) pull-down assay (Fig. 3c). Indeed, GCLM was shown to interact mainly with NKRF in the nucleus by Co-IP assays with subcellular fractions (Fig. 3d and Supplementary Fig. 3c), which was also verified by the overexpression of Flag-GCLM-NLS (Fig. 3e and Supplementary Fig. 3d) and Duolink assays (Fig. 3f). Moreover, the association of GCLM and NKRF was enhanced following platinum drug treatment in a dose-dependent manner and was not affected by mutation of the enzyme active sites in GCLM (Fig. 3g and Supplementary Fig. 3e). Furthermore, since the NKRF sequence is composed primarily of four main domains[20], we generated truncation and deletion mutants of NKRF to explore the regions involved in its interaction with GCLM (Supplementary Fig. 3f). The results revealed that the C-terminal fragment (M1) of NKRF was the main interaction region and that the deletion of the DSRM1 domain (M3) of NKRF strongly affected its ability to bind GCLM (Fig. 3h). In addition, the results of the cellular phenotype assays showed that silencing NKRF could partially reverse decreases in cell viability and ATP levels and increase in the apoptosis rate in HCT116 cells induced by GCLM inhibition (Supplementary Fig. 3g–i). Taken together, our data indicate that nuclear GCLM may perform a moonlighting function by interacting with the NKRF DSRM1 domain.

Previous studies have shown that NKRF inhibits NF-κB activity by binding to a specific negative regulatory element (NRE) in the promoters of certain NF-κB-responsive genes, such as *iNOS* and *IFN-β*[21]. Other studies have revealed that NKRF interacts with the p65 to

modulate the activity of NF-κB[20]. We thus sought to characterize the interaction of nuclear GCLM-NKRF with the NF-κB pathway further and showed that GCLM inhibition had no significant effect on the interaction between NKRF and the NRE on the promoters of *iNOS or IFN-β* (Supplementary Fig. 3j). As shown in Fig. 3i, with no change in the expression level of NKRF, GCLM inhibition enhanced the interaction of NKRF with p65 (Fig. 3i), which resulted in the inhibition of the p65-p50 interaction (Fig. 3j), a typical heterodimeric transcription factor in the canonical NF-κB pathway[22]. The results also showed that GCLM did not obviously interact with p65 (Supplementary Fig. 3k). These findings imply that nuclear GCLM interacts with NKRF to relieve the repressive effect of NKRF on p65/p50. Furthermore, the dual-luciferase assay of promoter activity showed that GCLM depletion significantly reduced the transcriptional activity of NF-κB/p65 and inhibited the expression of representative NF-κB/p65 downstream genes (*CCND1, XIAP, BCL2, BCL-xl* and *iNOS*) that related to cell growth and apoptosis (Fig. 3k, l and Supplementary Fig. 3l)[23]. These above effects were rescued by re-expressing nuclear GCLM, and the increased NF-κB/p65 activity induced by re-expressing nuclear GCLM could be reversed by over-expressing NKRF (Fig. 3k, l and Supplementary Fig. 3l). In addition, the increase in cell viability and decrease in the apoptosis rate of CRC cells induced by nuclear GCLM overexpression were reversed by silencing p65 with or without oxaliplatin treatment (Fig. 3m and Supplementary Fig. 3m). These findings suggest that nuclear GCLM performs a non-canonical function via the GCLM-NKRF interaction to facilitate NF-κB/p65 activity and contribute to chemoresistance.

## Platinum drugs promote GCLM nuclear localization via GCLM binding to importin α5

Classically, the nuclear import of proteins is frequently regulated by importin α/β family members that bind nuclear localization signals (NLSs, rich in basic amino acids) of cargo proteins, which can be inhibited by importazole (IPZ)[24]. In elucidating the mechanism by which platinum drug treatment promotes the nuclear translocation of GCLM, we identified a potential bipartite NLS at the N-terminus of GCLM (4-36: DSRAAKALLARARTLHLQTGNLLNWGRLRKKCP) predicted by the cNLS Mapper tool (Fig. 4a). After mutating the main basic amino acids (R/K) in the NLS of GCLM to alanine (A, GCLM-Mut) (Fig. 4a), we found that GCLM-Mut was barely translocated to the nucleus even after treatment with oxaliplatin, as well as after treatment with IPZ (Fig. 4b, c). Furthermore, the interaction of GCLM and NKRF, even upon induction by oxaliplatin treatment, was inhibited by GCLM-Mut overexpression or IPZ treatment compared with GCLM-WT overexpression (Fig. 4d and Supplementary Fig. 4a). Consistently, compared with rGCLM-WT overexpression, rGCLM-Mut overexpression significantly inhibited the transcriptional activity of NF-κB/p65 (Fig. 4e), reduced the expression of representative NF-κB/p65 downstream genes (Supplementary Fig. 4b), decreased the viability of CRC cells (Fig. 4f and Supplementary Fig. 4c), and increased the

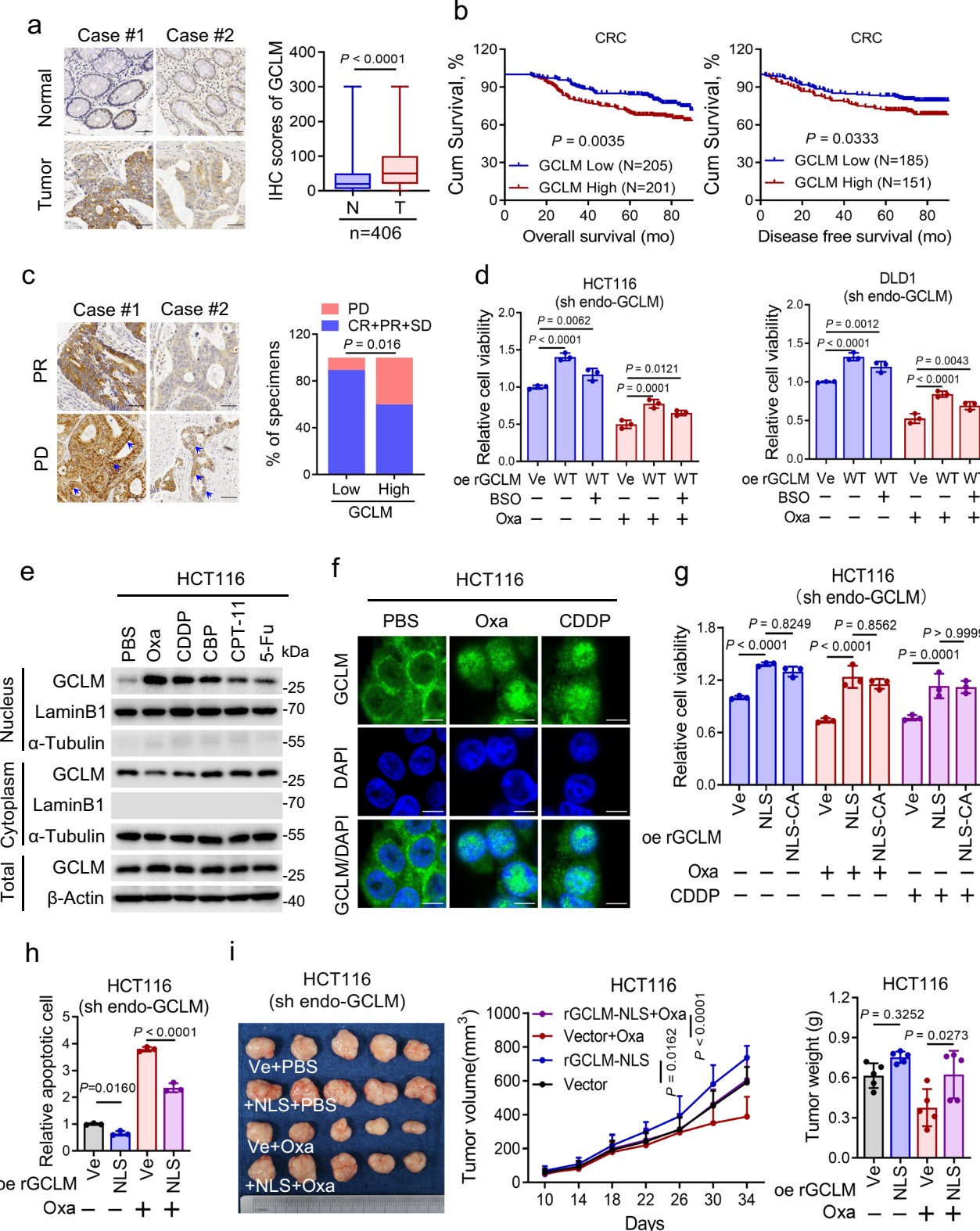

number of apoptotic cells in either the absence or presence of oxaliplatin treatment (Fig. 4g and Supplementary Fig. 4d). These results suggest that the accumulation of GCLM in the nucleus depends on its NLS.

Previous studies have shown that the importin α karyopherin functions as an adaptor that links to the NLS, and six importin α karyopherin family members (α1 and α3-7) have been identified in *human*[25]. We applied LC–MS/MS analysis to determine the specific importin α family member that interacts with the GCLM-NLS, and discovered that importin α5 (also known as KPNA1) was precipitated by GCLM in the presence of oxaliplatin (Supplementary Fig. 4e). The GCLM-importin α5 interaction was verified by Co-IP assays and immunofluorescence co-localization assays (Fig. 4h and Supplementary Fig. 4f), and this interaction was greatly enhanced by oxaliplatin treatment and reduced by mutation of the NLS of GCLM (Fig. 4i and Supplementary Fig. 4g). Consistently, silencing importin α5 reduced

**Fig. 2 | GCLM translocates to the nucleus up platinum drug treatment. a** IHC staining and scores of GCLM expression in primary CRC tumor (T) and adjacent normal tissues (N) (n = 406, CRC tissue specimens). Scale bar = 50 μm. The data are presented as a box-and-whisker graph (minimum–maximum), and the horizontal line across the box indicates the median (**a** right). **b** Overall survival and disease-free survival assays of CRC patients based on the GCLM protein level in (**a**). **c** IHC staining of GCLM expression in CRC patients receiving FOLFOX or XELOX chemotherapy. The blue arrowheads indicate the GCLM expression in nucleus. The correlation between GCLM expression and the response of patients to the standard chemotherapy (right), which is presented in the form of the percentage of total samples. (n = 58, CRC tissue specimens). Scale bar = 50 μm. **d** Cell viability of CRC cells overexpressing control or rGCLM WT, which were treated with PBS, oxaliplatin (40 μM for HCT116, 80 μM for DLD1, 24 h) or BSO (150 μM, 24 h). **e** IB detection of total, cytoplasmic and nuclear GCLM expression in HCT116 cells treated with 40 μM oxaliplatin (Oxa), cisplatin (CDDP) and carboplatin (CBP), and 20 μM irinotecan

(CPT-11) and 5-fluorouracil (5-FU) for 24 h. **f** IF staining showing the localization of GCLM in HCT116 cells with the treatment of PBS, oxaliplatin or cisplatin (40 μM, 24 h). Scale bar = 10 μm. Cell viability (**g**) and apoptotic cells (**h**) among HCT116 cells overexpressing control, nuclear GCLM (NLS) or C193/194 A mutant (NLS-CA) with oxaliplatin or cisplatin treatment (40 μM, 24 h). **i** Photographs and tumor volumes, weights analysis of CDX model after the implantation of HCT116 cells overexpressing control or nuclear GCLM, followed by intraperitoneal injections of PBS or oxaliplatin (5 mg/kg). IB experiments were repeated three times and n = 3 biologically independent experiments in (**d**, **g**, **h**) and n = 5 mice in (**i**). The data are presented as the mean ± S.D (**d**, **g**, **h**, **i**). The *P* values were calculated by two-tailed paired Student's *t* test (**a**), Kaplan–Meier analysis (log-rank test) (**b**), two-sided chi-square test (**c**), one-way ANOVA (**h**, **i** right), and two-way ANOVA (**d**, **g**, **i** middle). PD progressive disease, SD stable disease, PR partial response, CR complete response, NLS nuclear localization signals.

---

the nuclear accumulation of GCLM induced by oxaliplatin treatment in HCT116 and DLD1 cells (Fig. 4j, k and Supplementary Fig. 4h–j). In addition, inhibiting importin α5 obviously reversed the increase in NF-κB/p65 activity, cell viability, ATP levels, and the reduced rate of apoptosis induced by rGCLM overexpression in HCT116 cells treated with oxaliplatin (Fig. 4l, m and Supplementary Fig. 4k, l). Taken together, these results demonstrate that platinum drug treatment promotes GCLM nuclear localization via GCLM binding to importin α5 in an NLS-dependent manner.

## P38 MAPK-mediated phosphorylation of GCLM participates in the platinum drug-induced nuclear localization of GCLM

Because the total level of the GCLM protein was only slightly affected by platinum drug treatment (Fig. 2e and Supplementary Fig. 2d), we speculated that a posttranslational modification of GCLM may be involved in platinum drug-induced GCLM nuclear accumulation. Further screening showed that the level of threonine (T) phosphorylation of GCLM was significantly increased under platinum drug treatment, rather than the serine and tyrosine phosphorylation level or the ubiquitination, acetylation or methylation level (Fig. 5a and Supplementary Fig. 5a). In addition, the GCLM-NKRF and GCLM-importin α5 interactions were found to be blocked by calf intestinal phosphatase (CIP) treatment, which resulted in GCLM dephosphorylation (Fig. 5b). Next, we used the PhosphoSitePlus database to identify the specific phosphorylated threonine residues in GCLM may affected by platinum drug treatment, and identified three putative threonine phosphorylation sites, namely, T17, T106 and T164. We then generated corresponding antibodies specific for phosphorylation at these residues (Supplementary Fig. 5b). Notably, the level of T17 phosphorylation but not the level of T106 or T164 phosphorylation in GCLM was increased by either oxaliplatin or cisplatin treatment in CRC cells (Fig. 5c and Supplementary Fig. 5c), and the T17 residue in GCLM is evolutionarily conserved among several species (Supplementary Fig. 5d). Additionally, we mutated T17 to alanine (T17A) or glutamic acid (T17E) to mimic the constitutive dephosphorylation or phosphorylation of GCLM, respectively, which did not affect the GSH level, suggesting no change in the enzyme activity of GCLM (Supplementary Fig. 5e). The results showed that overexpression of the GCLM T17A mutant, but not the GCLM WT or the T17E mutant, significantly diminished the oxaliplatin-induced nuclear accumulation of GCLM (Fig. 5d, e). Mechanistically, compared with GCLM WT or GCLM T17E mutant overexpression, GCLM T17A mutant overexpression repressed the binding of GCLM to importin α5 and subsequently to NKRF upon oxaliplatin treatment (Fig. 5f). These data suggest that the platinum drug-induced GCLM-importin α5 interaction depends on the phosphorylation of GCLM at T17.

To further identify the kinase that mediates the phosphorylation of GCLM at T17 upon platinum drug stimulation, a panel of inhibitors linked to DNA damage-responsive kinase signaling (AKT, JNK, ERK, and

P38 MAPK) were utilized, because DNA damage is the main cause of the cytotoxicity associated with platinum drug treatment[26]. We found that the level of T17-phosphorylated GCLM and the formation of the GCLM-importin α5 complex were greatly decreased by treatment with the P38 MAPK inhibitor but not the other inhibitors (Fig. 5g). Moreover, the increases in the level of T17 phosphorylated GCLM, the GCLM-importin α5 interaction and the nuclear accumulation of GCLM induced by oxaliplatin treatment were attenuated by the P38 MAPK inhibitor (Fig. 5h, i). Consistent with a previous study showing that P38 MAPK signaling can be activated in an ATM-dependent manner or through other unestablished mechanisms in response to DNA damage[27], our results showed that P38 MAPK and ATM signaling were activated by oxaliplatin treatment (Supplementary Fig. 5f). Next, we verified the importance of GCLM phosphorylation at T17 by depleting endogenous GCLM and re-expressed shRNA-resistant rGCLM WT, T17A and T17E in CRC cells (Supplementary Fig. 5g). Compared with rGCLM WT re-expression, overexpression of rGCLM T17A, but not rGCLM T17E, reduced NF-κB/p65 activity in both the absence and presence of oxaliplatin (Fig. 5j). In addition, compared with re-expression of rGCLM WT, re-expression of rGCLM T17A, but not re-expression of rGCLM T17E, decreased cell viability, ATP levels and increased the apoptosis rate, and these effects were more significant following oxaliplatin treatment (Fig. 5k, l and Supplementary Fig. 5h–j). Collectively, these findings indicate that the increased T17 phosphorylation of GCLM and its subsequent nuclear localization upon platinum drug stimulation are mediated by P38 MAPK kinases.

## Phosphorylation of GCLM at T17 contributes to CRC chemoresistance in vivo

To verify the enhanced role of targeting GCLM in modulating the sensitivity of CRC to oxaliplatin in situ, we further established an azoxymethane/dextran sodium sulfate (AOM/DSS)-induced spontaneous CRC model. This model used mice with conditional knockout of intestinal Gclm induced by tamoxifen and mice expressing floxed alleles of Gclm under the control of the pVillin-Cre-ERT2 promoter (Supplementary Fig. 6a, b). After five tamoxifen treatments, Gclm was inducibly depleted in intestinal epithelial tissues (Supplementary Fig. 6c, d). Remarkably, conditional knockout of Gclm during the late stages in the AOM/DSS mouse model reduced the multiplicity of large adenomas and the total load of adenomas (Fig. 6a), with reductions in both the number and volume of tumors compared with those in WT mice, and the suppressive effects were more significant following oxaliplatin treatment (Fig. 6b and Supplementary Fig. 6e). H&E and immunohistochemical (IHC) staining also revealed that Gclm knockout mice presented less severe colon inflammation, a further decrease in cell proliferation (Ki67 staining) and increased cell death (Tunel assay) compared with WT mice (Fig. 6c, d and Supplementary Fig. 6f). Compared with either monotherapy, the combined group of GCLM knockout and oxaliplatin treatment resulted in superior antitumor

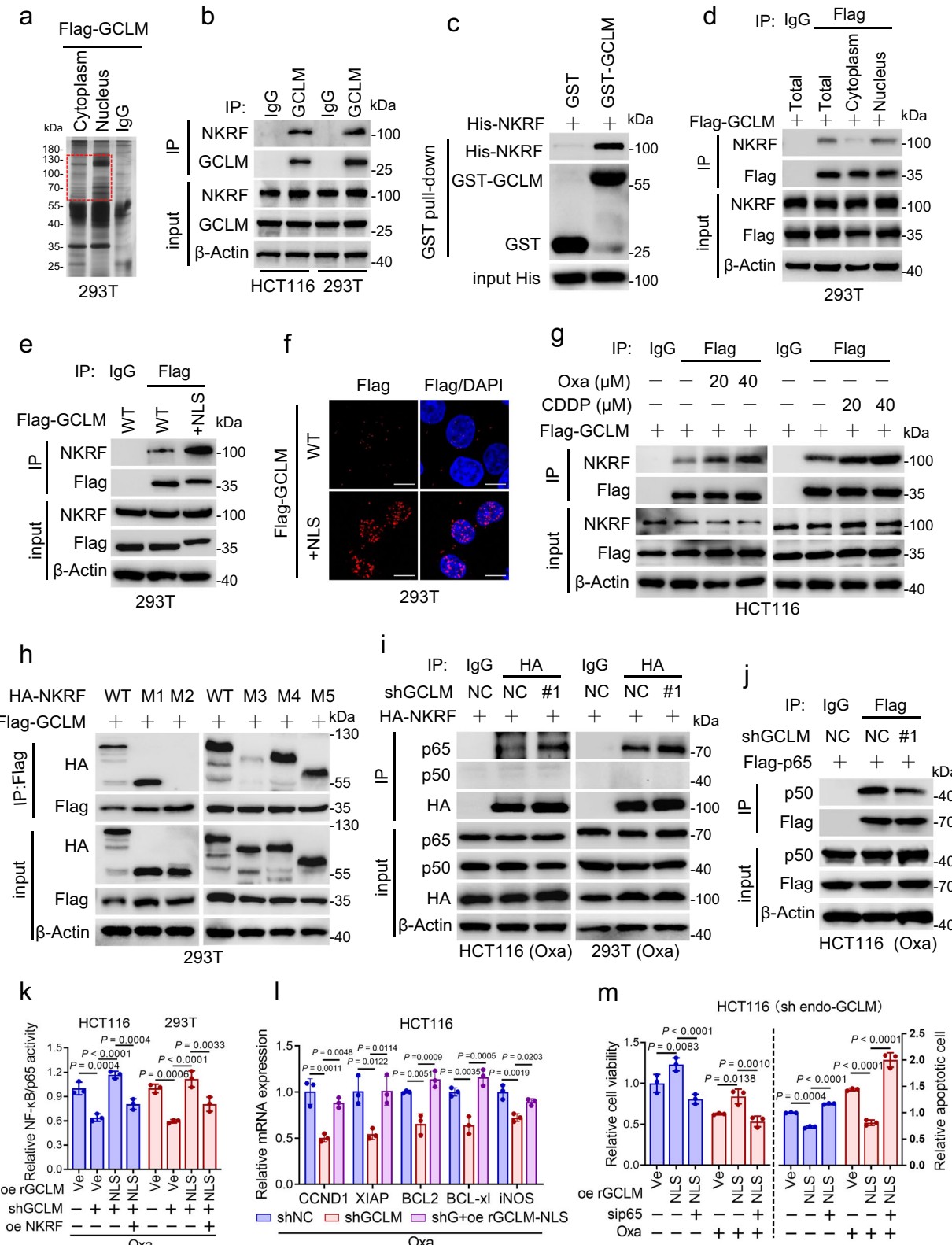

efficacy (Fig. 6a–d). Consistently, these results suggest that intestinal Gclm deficiency could enhance the chemotherapeutic effect of oxaliplatin in the context of AOM/DSS challenge.

HCT116 or DLD1 cells re-expressing rGCLM WT, T17A or T17E were implanted after the knockdown of endogenous GCLM to verify the importance of T17 phosphorylation in GCLM-induced chemoresistance in vivo. Compared with those in the rGCLM WT re-expression group, tumor growth and weights were decreased in the rGCLM T17A

re-expression group but not in the rGCLM T17E re-expression group in the absence or presence of oxaliplatin treatment (Fig. 6e, f and Supplementary Fig. 6g, h). In addition, we used P38 MAPK inhibitors targeting GCLM phosphorylation to assess their efficacy in sensitizing CRC tumors to the oxaliplatin-based clinical standards of care (FOLFOX regimen) in the PDX model. The results revealed that P38 MAPK inhibitor treatment decreased the tumor volume and weight, and the combination group of the P38 MAPK inhibitor and FOLFOX regime

**Fig. 3 | Nuclear GCLM interacts with NKRF to orchestrate NF-κB activity and chemoresistance. a** Coomassie blue staining showing the bound proteins of cytoplasmic and nuclear GCLM in 293 T cells overexpressing Flag-tagged GCLM under oxaliplatin treatment (40 μM, 24 h). Red box indicates the main proteins that differed between the cytoplasmic and nuclear fractions. **b** Co-IP analysis showing the level of NKRF bound by GCLM using anti-GCLM or IgG antibody. **c** GST pull-down analysis detecting the level of purified His-NKRF interacting with purified GST-GCLM. **d** Co-IP analysis demonstrating the level of NKRF bound by total, cytoplasmic and nuclear GCLM in 293 T cells overexpressing Flag-tagged GCLM. Co-IP (**e**) and Duolink (**f**) analyses showing the interaction (red dots in **f**) of GCLM and NKRF in 293 T cells overexpressing Flag-tagged GCLM WT or nuclear GCLM ( + NLS). Scale bar = 10 μm. **g** Co-IP analysis showing the interaction of GCLM and NKRF in HCT116 cells treated with different concentrations of oxaliplatin or cis-platin for 24 h. **h** Co-IP analysis demonstrating the level of GCLM bound by WT or truncated NKRF mutants (M1-M5) in 293 T cells. Control or GCLM-knockdown

HCT116 or 293 T cells overexpressed HA-tagged NKRF. Co-IP analysis showing the interaction of NKRF and p65/p50 (**i**) and the interaction of p65 and p50 (**j**) with oxaliplatin treatment (40 μM, 24 h). **k** HCT116 or 293 T cells were depleted GCLM and re-expressed nuclear GCLM (NLS) with control or NKRF inhibition in the presence of oxaliplatin (40 μM, 24 h). The transcriptional activity of NF-κB/p65 was detected. **l** Q-PCR analysis of the expression of NF-κB/p65-targeted genes (CCND1, XIAP, BCL2, BCL-xl, and iNOS) in HCT116 cells when depleting GCLM and re-expressing nuclear GCLM in the presence of oxaliplatin (40 μM, 24 h). **m** Cell viability and apoptotic cells analysis in HCT116 cells when depleting GCLM and re-expressiong nuclear GCLM with control or p65 silencing with or without oxaliplatin treatment (40 μM, 24 h). IB experiments were repeated three times and n = 3 biologically independent experiments in (**k–m**). All the data are presented as the mean ± S.D. The *P* values were calculated by one-way ANOVA (**k**, **l**) and two-way ANOVA (**m**).

treatment resulted in the greatest reductions in volume and weight in both PDX #1 and PDX #2 models (Fig. 6g, h and Supplementary Fig. 6i). Collectively, these data show that the phosphorylation of GCLM at T17 plays a crucial role in GCLM nuclear translocation and subsequent chemoresistance in CRC cells.

## Nuclear GCLM is highly expressed in CRC and indicates a poor prognosis

To investigate the clinical implications of our findings, we examined the levels of nuclear GCLM and T17-phosphorylated GCLM in CRC patients from SYSUCC (n = 406; related clinicopathological information is provided in Supplementary Table 1). The level of GCLM in the nucleus and T17-phosphorylated GCLM were increased in primary CRC tissues compared with adjacent normal tissues (Fig. 7a, b). We categorized the nuclear and T17-phosphorylated GCLM levels as low or high in comparison compared with the median value, and Kaplan–Meier survival analysis strikingly indicated that patients with high nuclear GCLM levels had unfavorable overall survival and disease-free survival, as did patients with high levels of T17-phosphorylated GCLM (Fig. 7c, d). Univariable and multivariable analyses also indicated that nuclear GCLM expression was an independent prognostic factor in CRC patients (Supplementary Table 2). Moreover, high levels of nuclear GCLM and T17-phosphorylated GCLM expression were also associated with a poor prognosis for patients with gastric carcinoma (GC) and esophageal squamous cell carcinoma (ESCC) from SYSUCC, although no notable statistical significance was observed in patients with ESCC, probably due to the insufficient sample size (Fig. 7e and Supplementary Fig. 7a, b). To validate the clinical correlation between nuclear/T17-phosphorylated GCLM expression and the response of patients with CRC to oxaliplatin-based chemotherapy (FOLFOX or XELOX regimens), we found that nuclear GCLM or T17-phosphorylated GCLM staining were increased in patients who exhibit a poorer benefit from standard chemotherapy (Fig. 7f). In addition, the nuclear GCLM expression level was positively correlated with the levels of T17-phosphorylated GCLM, the level of Ki67, and P38 MAPK activity and negatively correlated with the Tunel-positive rate, which was consistent with our molecular mechanism (Fig. 7g, h). Taken together, these data further indicate that nuclear GCLM is a promising prognostic indicator and a potential therapeutic target for improving the sensitivity of CRC to platinum-based chemotherapy.

## Discussion

As an oncogene with high expression in multiple cancer types, GCLM is closely associated with the drug resistance of tumor cells. For example, chemotherapeutic agents, such as paclitaxel and gemcitabine, increase the GCLM mRNA level, and inhibition of GCLM blocks the enrichment of breast cancer stem cells induced by chemotherapy and impairs tumor initiation[14]. Furthermore, chemosensitivity is more effectively increased in small cell lung cancer when GCLM is inhibited than when

GCLC is inhibited, suggesting that GCLM may be a more effective target for ameliorating chemoresistance[15], which is consistent with our CRISPR screening and verification results. In our study, high GCLM expression was found in CRC samples and indicated a poor response to chemotherapy. GCLM repression significantly enhanced the sensitivity of CRC cells to platinum-based chemotherapy in vitro and in vivo. In addition, GCLM and GCLC are generally believed to be cytoplasmic proteins. A study showed that GCLM displayed a perinuclear focal localization in hepatocytes with hepatocyte-specific deletion of GCLC or a low level of GSH[28]. However, the specific molecular mechanism of GCLM nuclear translocation and the functions of GCLM in the nucleus have not been elucidated. We unexpectedly found that GCLM was localized in the nucleus in the tissues of CRC patients who benefited less from standard chemotherapy, and that selectively overexpressing nuclear GCLM reduced the sensitivity of CRC cells to platinum-based chemotherapy independently of its enzyme activity. Therefore, our data imply a previously uncharacterized function of nuclear GCLM that contributes to chemoresistance in CRC cells.

Accumulating evidence has recently revealed that multiple metabolic enzymes exhibit unexpected activities and functions in addition to their established roles in supporting malignant transformation by immediately undergoing changes in their subcellular localization, especially nuclear translocation, in response to pathological stimuli[12]. For example, the nucleus-translocated α-ketoglutarate dehydrogenase (α-KGDH) complex, acetyl-CoA synthetase short-chain family member 2 (ACSS2) and fumarase locally produce succinyl-CoA, acetyl-CoA and fumarate, respectively. These molecules play instrumental roles in histone succinylation, acetylation and methylation, respectively, and result in gene expression and tumorigenesis[29–31]. Besides, the nucleus-accumulated pyruvate kinase M2 isoform (PKM2) can directly bind to β-catenin, hypoxia-inducible factor 1α (HIF1α), and CtBP-interacting protein (CtIP) and phosphorylate signal transducer and activator of transcription 3 (STAT3) to regulate downstream gene expression and then facilitate tumor development[32–35]. In addition, nuclear fructose-1,6-bisphosphatase 1 (FBP1) inhibits nuclear HIF function via a direct interaction with the HIF inhibitory domain[36], nuclear programmed death-ligand 1 (PD-L1) enhances the antitumor response to PD-1 blockade by governing immune response gene expression[37], and nuclear hexokinase 2 (HK2) maintains stemness by regulating chromatin openness[24]. In our study, we found that GCLM translocated to the nucleus in response to platinum drug treatment, and that nuclear GCLM competitively interacted with NKRF to facilitate NF-κB activity and subsequently mediate chemoresistance (Fig. 7i). Moreover, the expression of GCLM in the nucleus was increased in CRC tissues and associated with shorter survival of CRC and GC patients. High levels of nuclear GCLM also exhibit a poorer benefit of CRC patients from standard chemotherapy. Therefore, our findings reveal a moonlighting function of GCLM and suggest that targeting the nuclear GCLM–NKRF–NF-κB axis constitutes a selective

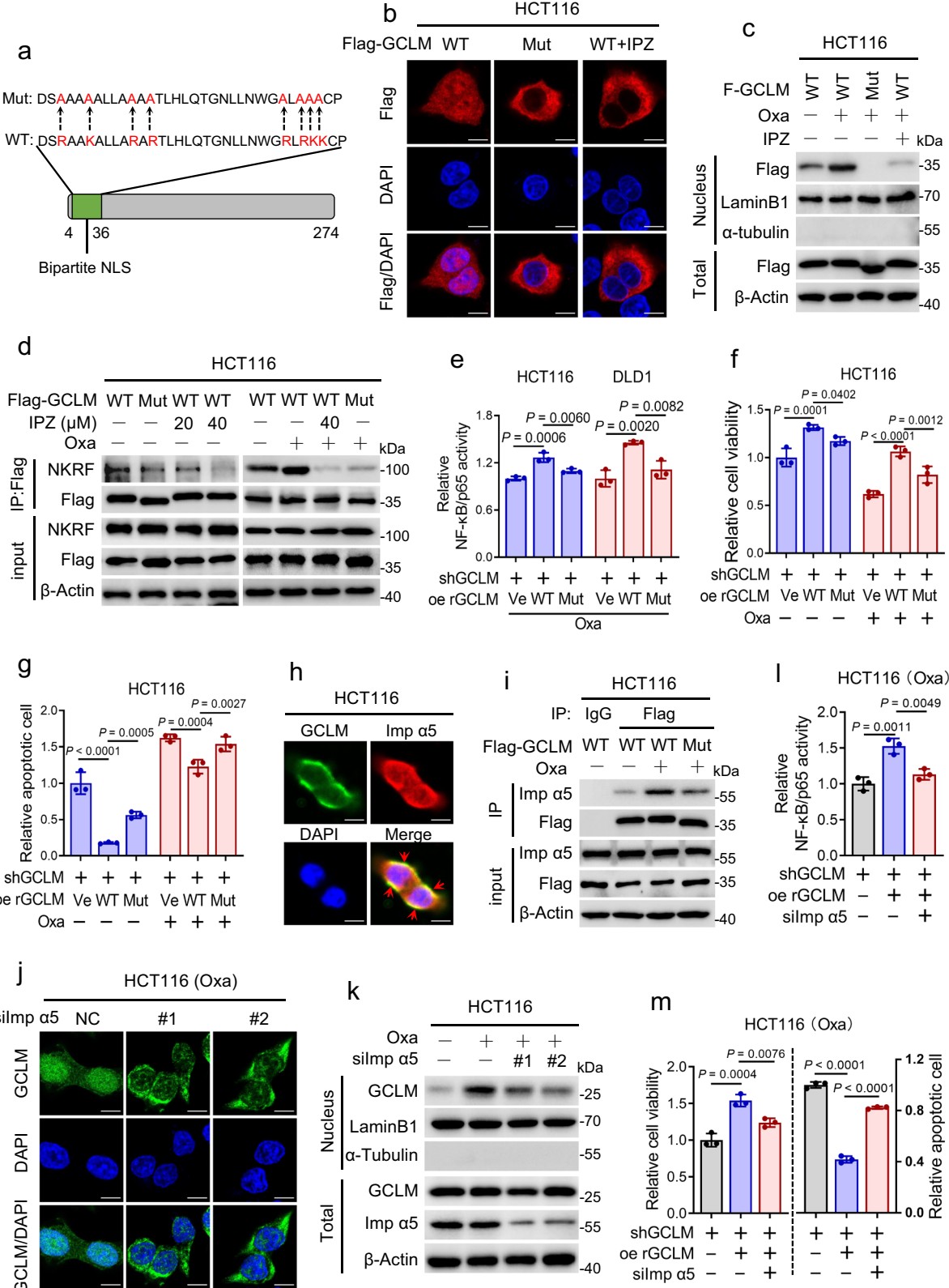

treatment for attenuating chemoresistance and CRC progression. Overall, understanding these noncanonical functions of metabolic enzymes can broaden the horizon of cancer metabolism research and lead to the discovery of therapeutic strategies in cancers. However, the crosstalk between the metabolic function and moonlighting function of GCLM and the roles of nuclear GCLM in other tumors and sensitivity to other chemotherapy drugs need to be further characterized.

The NF-κB pathway broadly promotes the transcription of target genes involved in multiple physiological and pathological processes, which consists of canonical and noncanonical pathways mediated by the p65/p50 and RelB/p52 heterodimers, respectively[22,23]. The canonical NF-κB pathway can be activated by various upstream signals induced by extracellular stimuli, including chemotherapeutic agents, to reduce antitumor responses, and inhibiting NF-κB or its upstream

**Fig. 4 | Platinum drugs promote GCLM nuclear localization via GCLM binding to importin α5. a** Schematic illustration of the potential NLS sequence of the GCLM protein predicted by the cNLS Mapper tool. The strategy for mutating the NLS of the GCLM is also shown. **b** HCT116 cells overexpressed Flag-tagged GCLM WT or GCLM-NLS mutant (Mut) with or without importazole treatment (40 μM, 24 h). The localization of GCLM was detected by IF staining. Scale bar = 10 μm. **c, d** HCT116 cells overexpressed Flag-tagged GCLM WT or GCLM-NLS Mut with oxaliplatin or IPZ treatment (40 μM, 24 h). IB detected the nuclear and total GCLM expression (**c**) and Co-IP analysis showed the level of interaction between NKRF and GCLM (**d**). **e–g** GCLM-knockdown HCT116 or DLD1 cells overexpressed rGCLM WT or rGCLM-NLS Mut with or without oxaliplatin treatment (40 μM, 24 h). The transcriptional activity of NF-κB/p65 (**e**), cell viability (**f**) and apoptotic cells (**g**) were detected. **h** Co-localization of endogenous GCLM and importin α5 in HCT116 cells was detected by IF. Red arrowheads showed the co-localization. Scale bar = 10 μm. **i** HCT116 cells overexpressed Flag-tagged GCLM WT or GCLM-NLS Mut with or without oxaliplatin treatment (40 μM, 24 h). Co-IP analysis showing the GCLM-importin α5 interaction. IF staining of the localization of GCLM (**j**) and IB detection of nuclear and total GCLM expression (**k**) in HCT116 cells with control or importin α5 silencing in presence of oxaliplatin treatment (40 μM, 24 h). Scale bar = 10 μm. **l, m** GCLM-knockdown HCT116 cells overexpressed vector or rGCLM WT with control or importin α5 silencing under oxaliplatin treatment (40 μM, 24 h). The transcriptional activity of NF-κB/p65 (**l**), cell viability and apoptotic cells (**m**) were detected. IB experiments were repeated three times and n = 3 biologically independent experiments in (**e–g, l, m**). All the data are presented as the mean ± S.D. The *P* values were calculated by one-way ANOVA (**e, l, m**) and two-way ANOVA (**f, g**). IPZ importazole, Imp α5 importin α5.

signaling can suppress tumor progression or chemoresistance[23]. For example, inhibition of the IL-1 receptor and TGF-β-activated kinase-1 (TAK1) promotes chemoresistance by abrogating NF-κB activation[38,39]. However, the underlying mechanism by which these chemotherapeutic agents regulate the NF-κB pathway remains to be further defined.

NKRF represses the transcriptional activity of NF-κB through multiple mechanisms. It was first found to interact with a specific negative regulatory element in the promoters of NF-κB target genes to repress their transcription via its DNA-binding ability[40]. Other evidence indicates that NKRF also directly interacts with NF-κB family members to inhibit the NF-κB-dependent gene expression, such as MYC, cyclin-dependent kinase 2 (CDK2) and solute carrier family 1 member 1 (SLC1A1)[20,41,42]. In addition, the highly expressed long noncoding RNA Uc003xsl.1 and miRNA-301a-3p directly bind to NKRF, disrupting its negative regulation of NF-κB-responsive genes to influence patient prognosis in triple-negative breast cancer and gastric cancer, respectively, and targeting these upstream mediators of NKRF could prevent tumor progression[43,44]. Consistent with these reports, we found that NKRF interacted with p65 to inhibit the p65-p50 interaction in the nucleus and then inhibited the transcription of apoptosis-related genes, which could be competitively rescued by GCLM-NKRF binding in the nucleus (Fig. 7i). Hence, these data replenish another regulatory mechanism of NF-κB signaling in response to platinum-based chemotherapy.

Protein phosphorylation is one of the most important post-translational modifications involved in metabolic reprogramming through its regulatory effects on protein stability, enzyme activity, or subcellular localization in response to stimuli and could be exploited to developing new therapeutic approaches for cancer[3,45]. For example, ATM and ERK2- mediated phosphorylation of PKM2 at T328 and S37, respectively, results in the accumulation of PKM2 in the nucleus, where it performs unexpected functions[25,32]. The phosphorylation of ACSS2 at S659 mediated by AMPK promotes the interaction of ACSS2 with importin α5, which also results in the nuclear translocation of ACSS2[29]. However, the role of GCLM phosphorylation has not been reported. In our study, we demonstrated that platinum drug treatment increases GCLM phosphorylation at T17 in a P38 MAPK activity-dependent manner, exposing the NLS of GCLM for importin α5 binding and subsequent nuclear translocation (Fig. 7i). In addition, an increased T17-phosphorylated GCLM level was observed in CRC tissues, and was associated with shorter survival of CRC and GC patients. High levels of T17-phosphorylated GCLM also exhibit a poorer benefit of CRC patients from standard chemotherapy. In addition, the level of GCLM in nucleus was positively correlated with the T17-phosphorylated GCLM level and P38 MAPK activity in CRC patients. These findings suggest a regulatory mechanism of GCLM and indicate that T17-phosphorylated GCLM may be a prognostic indicator for patients with CRC. However, we found that the GCLM phosphorylation at T17 was still present in the nucleus, suggesting that further studies are needed to investigate the other kinases that mediate the phosphorylation of GCLM.

In conclusion, our work reveals a moonlighting function of nuclear GCLM and implicates the P38 MAPK–nuclear GCLM–NF-κB axis as a prognostic and therapeutic target for CRC. In addition, selectively inhibiting the function of nuclear GCLM through the development of targeted inhibitors, which will be the focus of future studies, is a potential therapeutic strategy for improving the antitumor efficacy of platinum-based chemotherapy in CRC.

## Methods
### Ethical statement
All patient samples were obtained from Sun Yat-sen University Cancer Center (SYSUCC, Guangzhou, China). After written informed consent was obtained from the participants who provided samples, our study was approved by the Medical Ethics Committee of SYSUCC (G2021-029-01) and complied with the Declaration of Helsinki. All animal experiments were performed based on the protocol approved by Institutional Ethics Committee for Clinical Research and Animal Trials of the SYSUCC (L102012021001X), and all mice were housed in a temperature-controlled room under pathogen-free conditions on a 12 h light–dark cycle. Transplanted tumors were not to exceed a diameter of 2.0 cm or 10% of body weight as permitted by the Institutional Ethics Committee for Clinical Research and Animal Trials of the SYSUCC.

### Cell culture
The human embryonic kidney (HEK) 293 T, human HCT116, and DLD1 CRC cell lines were purchased from the American Type Culture Collection (ATCC, Manassas, VA, USA). According to standard protocols, 293 T cells were maintained in DMEM and other cells in RPMI 1640 (Thermo Fisher Scientific, Carlsbad, CA, USA) supplemented with 10% fetal bovine serum (ExCell Bio, Shanghai, China) and 1% penicillin/streptomycin (WISENT, Nanjing, China) at 37 °C with 5% $CO_2$. All cells were authenticated on the strength of short tandem repeat (STR) fingerprinting at the Medicine Lab of Forensic Medicine Department of Sun Yat-sen University (Guangzhou, China), and tested negatively for mycoplasma contamination before use.

### Human tissue specimens
We obtained 406 pairs of normal versus CRC tissue, 235 GC and 276 ESCC tissues specimens receiving surgery at SYSUCC, 58 CRC tissues with FOLFOX or XELOX chemotherapy (tissue samples are collected before treatment) for IHC staining analysis from SYSUCC[18]. All patients were treated with standard of care according to treatment guidelines and were followed up on a regular basis. The clinicopathological characteristics of these patients with CRC are summarized in Supplementary Tables 1 and 2.

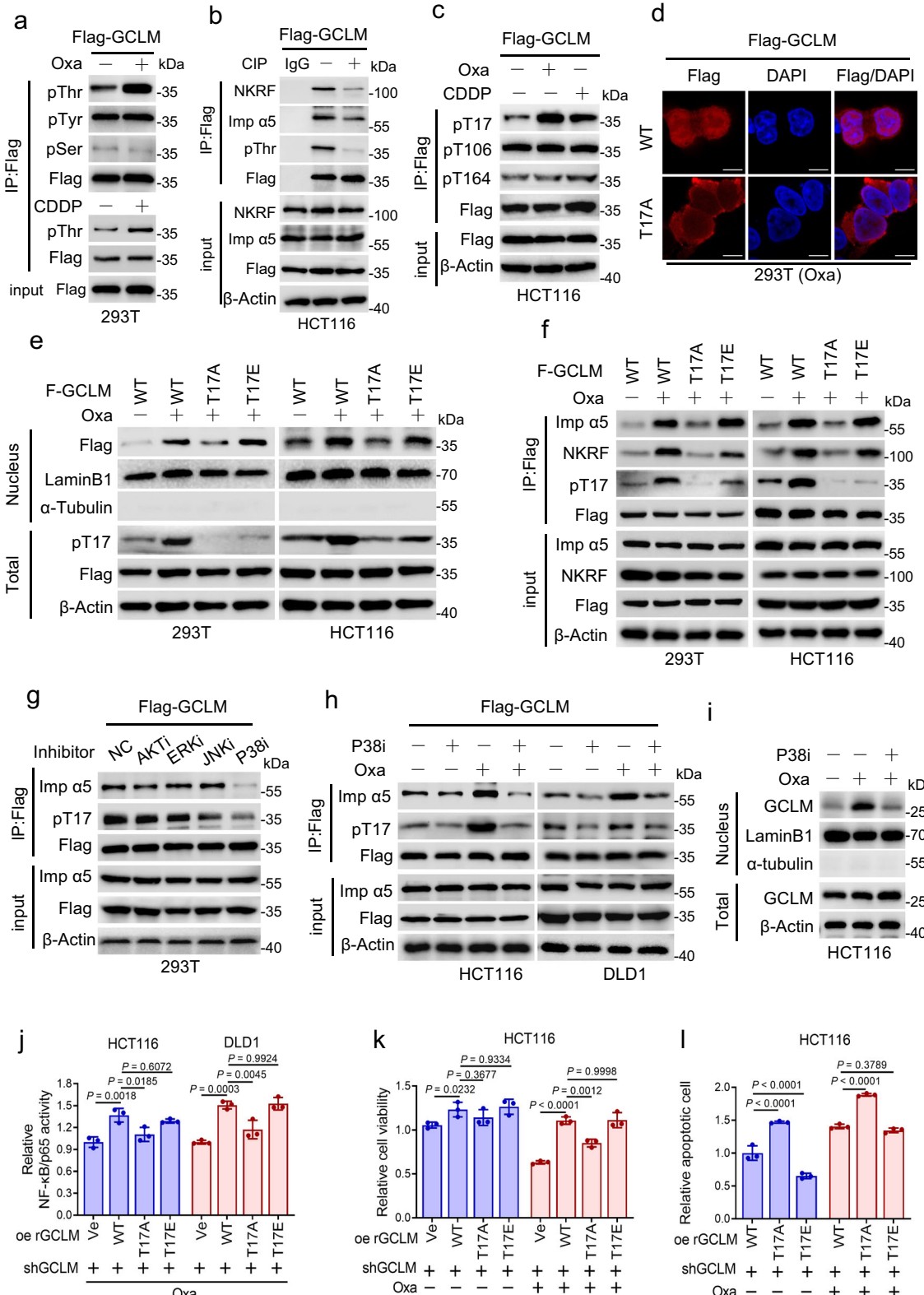

## CRISPR-Cas9 knockout screen

The human metabolic enzyme sgRNA library, which targets 1773 metabolic enzyme-encoding genes with 4 sgRNAs per gene, was designed and synthesized by GENEWIZ (Suzhou, China) and was then packaged into lentiviruses[46]. For the in vitro CRISPR-Cas9 knockout screen in the presence of oxaliplatin, $4 \times 10^7$ HCT116 cells were plated and transduced with the lentiviral library at a low multiplicity of infection (MOI of 0.3) with 10 μg/mL polybrene to ensure that most cells received only one viral construct with high probability. The infected cells were selected with puromycin (0.8 μg/mL) for 7 days[3]. The selected cells were treated with PBS or 10 μM oxaliplatin for 7 days. Thereafter, the remaining surviving cells were harvested, and their genomic DNA was isolated and amplified via 2-step PCR using NEBNext Ultra II Q5 Master Mix (M0544L, New England Biolabs, Ipswich, MA, USA) and the primers used are listed in Supplementary data 4. The PCR products containing the sgRNA sequences were extracted from the

**Fig. 5 | P38 MAPK-mediated phosphorylation of GCLM participates in platinum drug-induced nuclear localization of GCLM. a** IP analysis showing the level of GCLM phosphorylated at threonine (pThr), serine (pSer) and tyrosine (pTyr) residues in 293 T cells treated with oxaliplatin or cisplatin (40 μM, 24 h). **b** Co-IP analysis showing the level of NKRF and importin α5 bound by GCLM in HCT116 cells treated with calf intestinal phosphatase (CIP) that resulted in GCLM dephosphorylation. **c** IP analysis showing the level of GCLM phosphorylated at Thr17, Thr106 and Thr164 in HCT116 cells treated with oxaliplatin or cisplatin (40 μM, 24 h). **d** IF staining showing the localization of Flag-GCLM in 293 T cells overexpressing Flag-tagged GCLM WT or T17A mutant with oxaliplatin treatment (40 μM, 24 h). Scale bar = 10 μm. **e, f** 293 T and HCT116 cells overexpressed the Flag-tagged GCLM WT or T17A, T17E mutant with or without oxaliplatin treatment (40 μM, 24 h). IB detection of nuclear and total GCLM expression (**e**) and Co-IP analysis showing the levels of importin α5 and NKRF interacting with GCLM (**f**).

**g** 293 T cells were treated with the inhibitors of AKT (AKTi, MK-2206, 5 μM), ERK (ERKi, PD98059, 10 μM), JNK (JNKi, SP600125, 20 μM) or P38 MAPK (P38i, SB203580, 10 μM) for 24 h. Co-IP analysis showing the GCLM-importin α5 interaction and the level of GCLM phosphorylated at Thr17. **h, i** CRC cells were treated with control or P38 MAPK inhibitor (P38i, SB203580, 10 μM) in the presence or absence of oxaliplatin treatment (40 μM for HCT116, 80 μM for DLD1, 24 h). Co-IP analysis showing the level of importin α5 interacting with GCLM (**h**) and IB detection of nuclear and total GCLM expression (**i**). The transcriptional activity of NF-κB/p65 (**j**), cell viability (**k**) and apoptotic cells (**l**) in GCLM-knockdown HCT116 or DLD1 cells overexpressing rGCLM WT or T17A, T17E mutants with or without oxaliplatin treatment (40 μM, 24 h). IB experiments were repeated three times and n = 3 biologically independent experiments in (**j–l**). All the data are presented as the mean ± S.D. The *P* values were calculated by one-way ANOVA (**j**) and two-way ANOVA (**k**, **l**).

gel, quantified, sequenced, and analyzed[3]. The CRISPR screen data are provided in Supplementary Data 1.

### RNA interference, lentivirus and plasmid transfection
Small interfering RNAs (siRNAs) targeting NKRF, importin α5 and p65 were synthesized by RiboBio (Guangzhou, China) and transfected using Lipofectamine RNAiMix (Thermo Fisher Scientific, Carlsbad, CA, USA). Lentiviruses packaging GCLM shRNAs and the CRISPR-Cas9 library were synthesized by OBiO Technology (Shanghai, China). CRC cells were transfected with lentiviruses and selected with puromycin (MedChemExpress, NJ, USA) for 7 days. The sequences of the siRNAs, sgRNAs or shRNAs used are listed in Supplementary data 4. Flag-tagged plasmids for WT GCLM, GCLM-NLS, GCLM mutants, p65 and HA-tagged plasmids for WT NKRF, and truncated mutants were provided by Synbio Technologies (Jiangsu, China). The plasmids were transfected into the cells using ViaFect Transfection Reagent (Promega, Madison, WI, USA).

### RNA isolation and Q-PCR analysis
Total RNA was extracted using TRIzol Reagent (Thermo Scientific, Carlsbad, USA) and then reverse transcribed to cDNA using the Prime Script RT Master Mix Kit (Takara, Tokyo, Japan) according to the manufacturer's instructions. The resulting cDNAs were analyzed by Q-PCR using an GoTaq qPCR Master Mix (Promega, Madison, USA) in a LightCycler 480 instrument[47] (Roche Diagnostics, Switzerland), and the data were normalized to the expression of β-Actin, as an endogenous control. Relative gene expression was analyzed using the $2^{-\Delta\Delta Ct}$ method. The primers used in this study were synthesized by TSINGKE Biological Technology (Guangzhou, China) and are listed in Supplementary Data 4.

### Cell viability, ATP, GSH and apoptosis assays
CRC cells were seeded in 96-well plates (NEST Biotechnology, Jiangsu, China) overnight and then treated with corresponded drugs. Cell viability, ATP level, GSH level were analyzed with a CellTiter 96® AQueous One Solution Cell Proliferation Assay (MTS) kit (Promega, Madison, USA), CellTiter-Glo Luminescent Cell Viability Assay Kit (Promega, Madison, USA) and GSH/GSSG-Glo™ Assay Kit (Promega, Madison, USA) according to the manufacturer's instructions[3,6]. For MTS, the absorbance of 96-well plates at a wavelength of 490 nm was measured using a Synergy™ Multi-Mode Microplate Reader (BioTek Instruments, VT, USA). For ATP and GSH level, luminescence was recorded using Multifunctional Microplate Reader (Tecan, Switzerland). Apoptosis was analyzed with a Annexin V/PI Kit (KeyGEN, Nanjing, China) and conducted with a flow cytometer (Beckman Coulter, CA, USA). The gating strategy used for apoptosis assays is showed in Supplementary Fig. 8.

### Subcutaneous tumor model
Our experimental design did not involve sex or gender-related factors. The BALB/c nude mice (6 to 8 weeks old, half male and half female) used in CDX models and PDX models were obtained from Beijing Vital River Laboratory (Beijing, China). Our preliminary experiments showed that GCLM knockdown phenotypes were identical in male and female mice and only female mice were used for the following experiments to control variables. For CDX models, $2 \times 10^6$ GCLM-knockdown, rGCLM-overexpressing (WT, GCLM-NLS, T17A or T17E), or control CRC cells were injected subcutaneously into the flanks of each mouse. The mice were then treated with intraperitoneal (i.p.) injections of PBS or oxaliplatin (5 mg/kg twice a week) when the tumor volume reached 50–100 mm³. For CDX models, The fresh tumor samples from two patients with CRC were initially generated PDX models (Supplementary Fig. 1g)[16]. Tumor samples were subcutaneously implanted into the dorsal flanks of mice as the first generation (F0). Once an appropriate volume was reached, the tumors were excised, divided into equal pieces, and subcutaneously implanted into nude mice as the second generation (F1). When the tumors became palpable (50–100 mm³), the tumor-bearing mice were divided randomly into four groups. GCLM siRNA or control siRNA was intratumorally injected into mice, and the mice were then treated with oxaliplatin, FOLFOX, P38 MAPK inhibitor (SB203580) or PBS by i.p. injection. Oxaliplatin (5 mg/kg) and siRNAs (5 nmol per injection) were injected twice a week. P38 MAPK inhibitor (5 mg/kg) was injected twice a week and FOLFOX (oxaliplatin 5 mg/kg, 5-fluorouracil 25 mg/kg) was injected every 2 days. The longest diameter and width of the tumors from CDX and PDX mice were measured every 4 or 5 days and used to calculate the tumor volume using the equation $V = 0.5 \times D \times W^2$ (V, volume; D, diameter; and W, width). All mice were sacrificed after being treated with oxaliplatin 7 treatments, and the tumors were excised, photographed, weighed, and embedded in paraffin for further pathological analysis.

### Mouse model of AOM/DSS-induced spontaneous mouse CRC
The formation of spontaneous colon tumors was induced by AOM/DSS[8] (Supplementary Fig. 6b). In brief, the C57BL/6 J background mice (6 to 8 weeks old, half male and half female) with intestine-specific Gclm knockout (Gclm^iKO) were generated by crossing mice carrying a Gclm exon 1 floxed allele with Villin-Cre-ERT2 mice purchased from GemPharmatech (Guangdong, China) and intraperitoneally injected with 10 mg/kg AOM on day 1. After one week, the mice were administered 2% DSS in the drinking water for 7 days, and regular water was then provided for 14 days. This cycle was repeated 3 times. For inducible intestinal Gclm deletion after tumor development, 55 days after the AOM treatment, mice were intragastrically administered tamoxifen (150 mg/kg) for 5 days. Simultaneously, the mice were treated with PBS or oxaliplatin (5 mg/kg twice a week) by i.p. injection for 1 month after the last cycle of AOM/DSS administration to determine whether Gclm

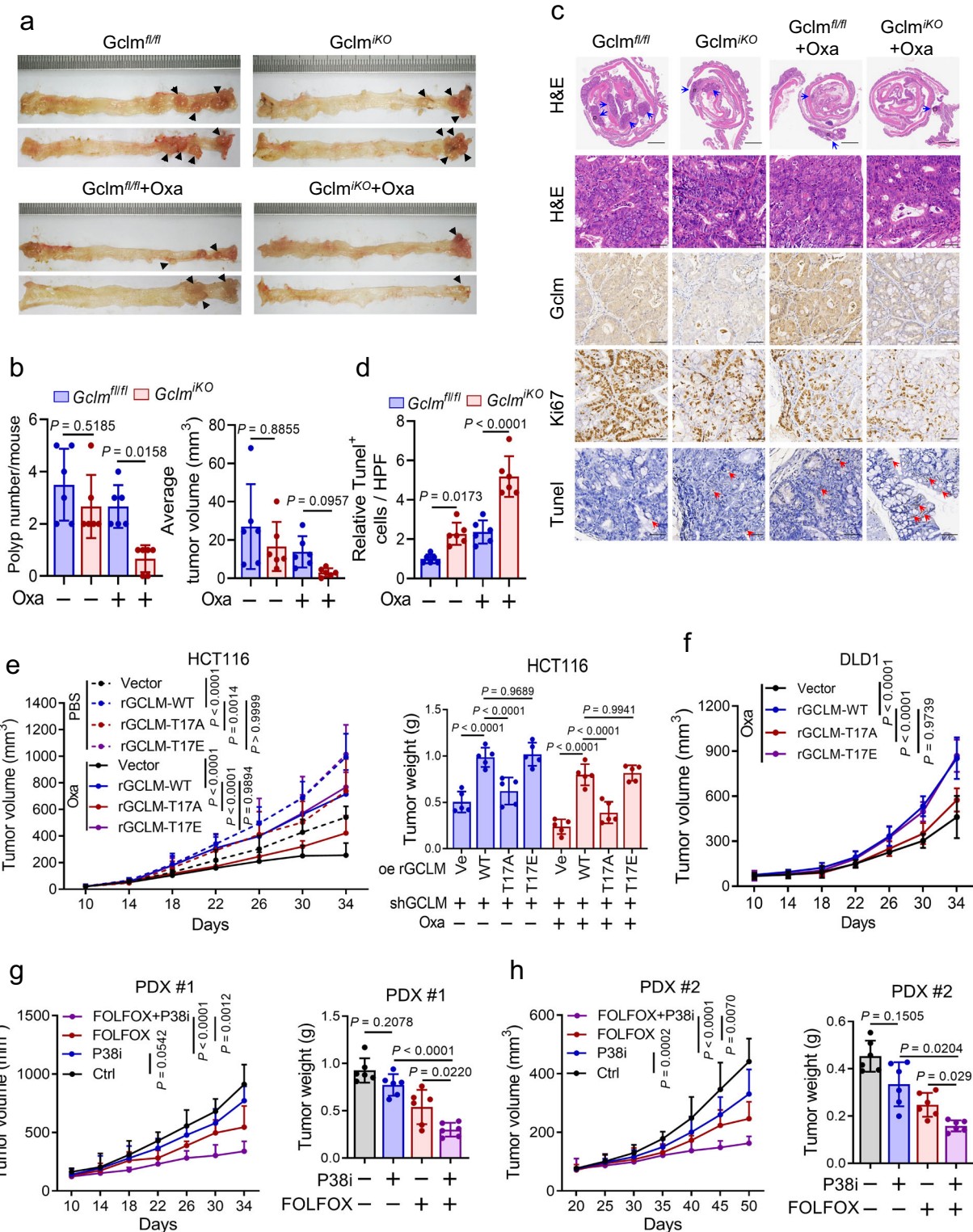

**Fig. 6 | Phosphorylation of GCLM at T17 contributes to CRC chemoresistance in vivo.** Representative images (**a**, the black arrowheads indicate the tumor), tumor numbers and average sizes (**b**) of spontaneous tumors in Gclm^{fl/fl} and Gclm^{iKO} mice treated with PBS or oxaliplatin (5 mg/kg). Representative images of H&E, IHC of GCLM, Ki67 and Tunel staining (**c**), and quantification of Tunel staining (**d**) of the spontaneous mouse CRC model. The blue arrowheads indicate the adenomas and the red arrowheads indicate the positive cells of Tunel staining. Scale bar = 1 mm (low power image) and 50 μm (high power images). Statistical analysis of the CDX tumor volume and weight after the implantation of endogenous GCLM-knockdown

HCT116 (**e**) and DLD1 (**f**) cells (2 × 10^6), which overexpressed rGCLM WT or T17A, T17E mutants, followed by intraperitoneal injections of PBS or oxaliplatin (5 mg/kg). Statistical analysis of the tumor volumes and weights in the PDX #1 (**g**) and PDX #2 (**h**) models, followed by intraperitoneal injections of control or P38 inhibitor (P38i, SB203580, 5 mg/kg) and FOLFOX (oxaliplatin 5 mg/kg, 5-fluorouracil 25 mg/kg). n = 6 mice in (**b**, **d**, **g**, **h**) and n = 5 mice in (**e**, **f**). All the data are presented as the mean ± S.D. The *P* values were calculated by one-way ANOVA (**b**, **d** and **g**, **h** right) and two-way ANOVA (**e**–**h** left).

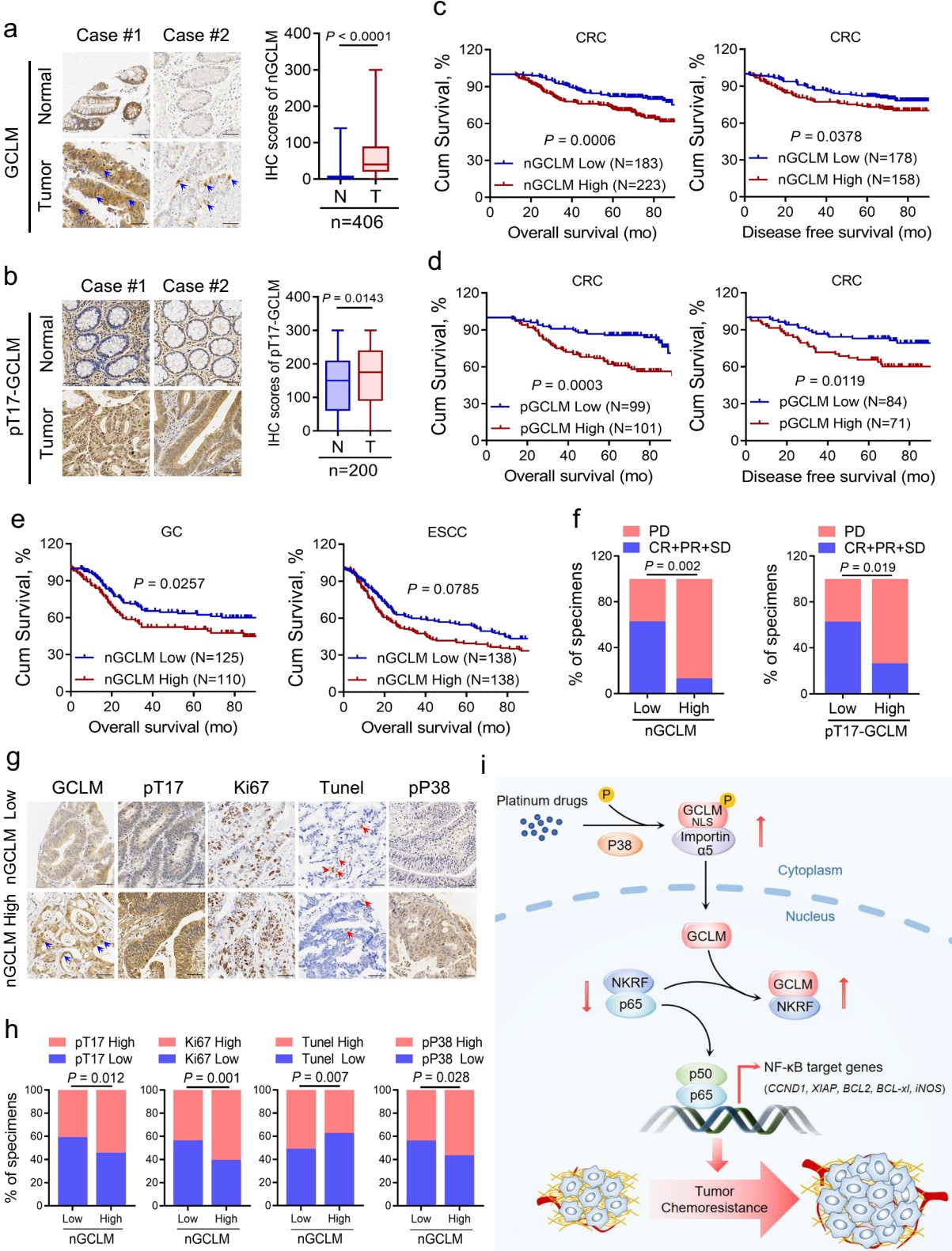

is constitutively required for tumor progression after tumor initiation. The animals were sacrificed on day 100 for general inspection and histological analysis. The knockout efficiency of Gclm knockout was verified by agarose gel electrophoresis and Q-PCR analysis of intestinal tissues. Tumors on the intestines were collected and evaluated to calculate the tumor volume using the following equation:

$V = 0.5 \times D \times W^2$. The tumors were excised, photographed, and embedded in paraffin for further pathological analysis.

## Subcellular fractionation

Cytoplasmic and nuclear fractions were isolated using a Nuclear and Cytoplasmic Extraction Kit (Thermo Fisher Scientific, Carlsbad, CA,

**Fig. 7 | Nuclear GCLM is highly expressed in CRC and indicates a poor prognosis. a** IHC staining and scores for nuclear GCLM expression (nGCLM, only focusing on GCLM staining in the nucleus, as shown by blue arrowheads) in paired primary CRC tumor tissues (T) and adjacent normal tissues (N) (n = 406, CRC tissue specimens). Scale bar = 50 μm. **b** IHC staining and scores for the level of GCLM phosphorylated at Thr17 (pT17-GCLM) in paired primary CRC tumor tissues (T) and adjacent normal tissues (N) (n = 200, CRC tissue specimens). Scale bar = 50 μm. **c**, **d** Overall survival (left) and disease-free survival (right) assays of patients with CRC based on nuclear GCLM (**a**) and pT17-GCLM (**b**) expression. **e** Overall survival assays of patients with GC (n = 235, GC tissue specimens) or ESCC (n = 276, ESCC tissue specimens) based on nuclear GCLM expression. **f** The correlation between nuclear GCLM and pT17-GCLM expression and the response of CRC patients to FOLFOX or XELOX chemotherapy (n = 58, CRC tissue specimens, PD progressive disease, SD stable disease, PR partial response, CR complete response). **g** IHC staining showing high and low expression of nuclear GCLM (nGCLM, blue arrowheads), pT17-GCLM, Ki67, Tunel (red arrowheads) and phospho-P38 MAPK (pP38) expression in CRC tumor tissues. Scale bar = 50 μm. **h** Correlations between nuclear GCLM expression and pT17-GCLM, Ki67, Tunel and pP38 MAPK expression. **i** Proposed working model based on this study. The model shows that P38 MAPK-mediated phosphorylated GCLM interacts with importin α5 and is translocated into the nucleus upon platinum drug treatment. Nuclear GCLM enhances NF-κB/p65 activity by interacting with NKRF to confer resistance to platinum-based chemotherapy in CRC cells. The data are presented as a box-and-whisker graph (minimum–maximum), and the horizontal line across the box indicates the median (**a**, **b** right), and the percentage of total sample (**f**, **h**). We categorized proteins levels as low or high compared with the median value of IHC score (**c**–**e**, **f** and **h**). The *P* values were calculated by two-tailed paired Student's *t* test (**a**, **b** right), Kaplan–Meier analysis (log-rank test) (**c**–**e**), and two-sided chi-square test (**f**, **h**). GC, gastric carcinoma; ESCC, esophageal squamous cell carcinoma.

USA) according to the manufacturer's instructions. Cytoplasmic and nuclear proteins were used in immunoblotting and IP assays.

## GST pull-down assay
GST pull-down assays were performed using Glutathione Sepharose® 4B beads (GE Healthcare, Little Chalfont, UK). First, 100 μl of GST beads was washed with PBST three times, and 20 μg of purified GST-GCLM protein or GST control (Genecreate, Wuhan, China) was then added and incubated for 1 h at 4 °C. Purified His-NKRF (10 μg) protein (Genecreate, Wuhan, China) was incubated with the GST-GCLM-bead complex for 2 h at 4 °C. After the incubation, the proteins were washed with PBST five times, eluted by boiling in 1× loading buffer and analyzed by immunoblotting.

## Duolink in situ proximity ligation assay (PLA)
The interaction of nuclear GCLM and NKRF was visualized according to the manufacturer's instructions for the Duolink In Situ Detection Reagents Red Kit (Sigma-Aldrich, MO, USA) as previously described[6]. In brief, cells were plated, fixed, permeabilized, and blocked with Duolink® Blocking Solution. The primary antibodies (anti-Flag-GCLM and anti-NKRF), secondary PLA probes and DAPI were added to the cells. Positive reactions were detected with a 594 nm fluorescence label detection kit. Hybridization between the two PLA plus and minus probes leading to fluorescent signal only occurs when the distance between the two antigens is less than 40 nm. Images of the cells were obtained using an LSM880 microscope with Fast Airyscan (Zeiss, Germany).

## NF-κB p65 transcription factor activity assay
The DNA binding activity of NF-κB/p65 was evaluated using the Dual-Luciferase Reporter Assay System (Promega, Madison, WI, USA). In brief, 293 T or CRC cells were transiently transfected with the Renilla luciferase (internal control) or NF-κB luciferase reporter plasmid (Beyotime, Jiangsu, China), and firefly and Renilla luminescence were recorded using Multifunctional Microplate Reader (Tecan, Switzerland)[48]. For each sample, luciferase activity was normalized to renilla luciferase activity.

## Immunoblotting (IB), Co-Immunoprecipitation (IP)
Proteins were extracted, separated, transferred to PVDF membranes (Bio-Rad Laboratories, Hercules, CA, USA), incubated with primary and secondary antibodies and visualized via a chemiluminescence assay (Thermo Fisher Scientific, Carlsbad, CA, USA)[3]. β-Actin, LaminB1 and α-Tubulin were included as loading controls for total, nuclear and cytoplasmic protein, respectively. Unprocessed images of the immunoblots are provided in the Source data. For Co-IP, the total cell lysate or cytoplasmic and nuclear fractions were incubated with anti-Flag, HA, GCLM, NKRF antibody or IgG and protein A/G beads (MedChemExpress, NJ, USA) overnight at 4 °C. The protein-magnetic bead complexes were washed and eluted with 1× loading buffer, and the bound proteins were subjected to IB or coomassie blue staining analysis for further LC-MS/MS analysis[6]. The antibodies used in this study were listed in Supplementary Data 4.

## Immunohistochemistry (IHC) assay
Paraffin-embedded sections were deparaffinized with xylene and rehydrated with 100% alcohol, 95% alcohol, 90% alcohol, 80% alcohol and 70% alcohol sequentially. Endogenous peroxidase activity was blocked with 3% hydrogen peroxide for 10 min. Antigen retrieval was performed using sodium citrate buffer or EDTA for 10 min at a sub-boiling temperature. Samples were blocked with 10% FBS for 1 h at room temperature, incubated with the primary antibody at 4 °C overnight and followed with a biotinylated secondary antibody for 1 h at room temperature. The color was developed using a Dako REAL™ EnVision™ Detection System (Copenhagen, Denmark), and counterstaining of nuclei was performed using hematoxylin. The stained sections were reviewed and scored independently based on their intensity: 0, 1, 2, and 3. The total score was obtained by multiplying the staining intensity score by the percentage of cells with positive staining. The score for nuclear GCLM focuses only on the staining of GCLM in nucleus.

## Mass spectrometry analysis of GCLM-interactome
Mass spectrometry analysis were performed by FitGene Biotechnology Co., Ltd. (Guangzhou,China). The total cell lysate or cytoplasmic and nuclear fractions from 293 T cells overexpressing Flag-GCLM were used to immunoprecipitation assay with anti-FLAG or IgG antibody. Protease digestion of the bound proteins solution was performed with TCEP, MMTS and IAA in UA buffer to block reduced cysteine. Finally, the protein suspension was digested with 2% trypsin overnight at 37 °C. The peptide was collected by centrifugation at 12,000 g for 10 min and vacuum dry at low-temperature.

LC-MS/MS experiments were performed on a Q Exactive hybrid quadrupole-Orbitrap mass spectrometer (ThermoFisher Scientific). Peptides were dissolved in 0.1% FA and 2% ACN, directly loaded onto a reversed-phase analytical column (75 μm i.d. x 150 mm, packed with Acclaim PepMap RSLC C18, 2 μm, 100 Å, nanoViper). The gradient was comprised of an increase from 5% to 50% solvent B (0.1% FA in 80% ACN) over 40 min, and climbing to 90% in 5 min, then holding at 90% for the 5 min. All at a constant flow rate of 300 nl/min. The peptides were subjected to NSI source followed by tandem mass spectrometry (MS/MS) in Q ExactiveTM coupled online to the UPLC. Intact peptides were detected in the Orbitrap at a resolution of 70,000. Peptides were selected for MS/MS using NCE setting as 27; ion fragments were detected in the Orbitrap at a resolution of 17,500. A data-dependent procedure that alternated between one MS scan followed by 20 MS/MS scans was applied for the top 20 precursor ions above a threshold ion count of 1E4 in the MS survey scan with 30.0 s dynamic exclusion. The electrospray voltage applied was 2.0 kV. Automatic gain control (AGC)

was used to prevent overfilling of the ion trap; 1E5 ions were accumulated for generation of MS/MS spectra. For MS scans, the m/z scan range was 350 to 1800 m/z. Fixed first mass was set as 100 m/z.

Protein identification were performed with MASCOT (http://www.matrixscience.com/) software by searching Uniprot_Aedis Aegypti. Searching Parameters are as follows: Fixed modifications: Carbamidomethyl (C); Variable modifications: Oxidation (M); Enzyme: Trypsin; Maximum Missed Cleavages: 1; Peptide Mass Tolerance: 20ppm; Fragment Mass Tolerance: 0.6 Da; Mass values: Monoisotopic; Significance threshold: 0.05. The LC-MS/MS analysis data are provided in Supplementary Data 2 and 3. The mass spectrometry raw data have been deposited to the ProteomeXchange Consortium via the PRIDE partner repository with the dataset identifier PXD055376.

### Masson's trichrome staining
Masson's trichrome staining analysis with Servicebio® Masson trichrome staining Kit (Servicebio Technology, Wuhan, China) according to the manufacturer's instructions. Briefly, Paraffin-embedded sections were deparaffinized and rehydrated. Soak the sections in 2.5% potassium bichromate solution overnight at room temperature and 30 min at 65 °C and wash in distilled water. Then, the sections were stained with Weigert's iron hematoxylin solution for 3 min and differentiated in 1% hydrochloric acid alcohol, stained with the Beibrich-Scarlet Acid Fuschin solution for 6 min, differentiated in the phosphomolybdic-phosphotungstic acid solution for 2 min, stained with aniline blue solution for 30 s, differentiated with 1% acetic acid solution for 7 s three time, and quickly dehydrated through 95% ethyl alcohol and cleared in xylene. The collagen fibers stain blue with a red background.

### Immunofluorescence (IF) assay
Cells were plated on glass-bottom cell culture dishes (NEST Biotechnology, Jiangsu, China), fixed with 4% paraformaldehyde for 15 min at 37 °C and permeabilized with 0.2% Triton X-100 in PBS for 10 min at room temperature. Samples were blocked with 5% FBS for 1 h at room temperature, incubated with primary and fluorescence dye-conjugated secondary antibodies for overnight at 4 °C and 1 h at room temperature (green: goat anti-rabbit IgG (H + L), Alexa Fluor 488 nm conjugate; red: goat anti-mouse IgG (H + L), Alexa Fluor 594 nm conjugate). 4',6-diamidino-2-phenylindole (DAPI) (Invitrogen, Carlsbad, CA, USA) was then used for counterstaining the nucleus, and immunofluorescence microscopy images were obtained and analyzed using an LSM880 with Fast Airyscan (Zeiss, Germany).

### Chromatin immunoprecipitation (ChIP) assay
ChIP assays were performed using a Pierce magnetic ChIP Kit (Thermo Fisher Scientific, Carlsbad, CA, USA) according to the manufacturer's instructions. Briefly, $1 \times 10^7$ HCT116 cells were harvested and incubated with 27 μL of 37% formaldehyde for crosslinking. Thereafter, glycine was added to stop the reaction. The cross-linked cells were washed twice with PBS and lysed with 200 μL lysis buffer. Next, the cell lysate was sonicated on ice and the chromatin was sheared into 200–500 bp fragments. Then, the sheared DNA mixture was subjected to IP with 1 μg NKRF antibody or normal rabbit IgG-magnetic beads complexes overnight at 4 °C. The protein–DNA complexes were washed with IP buffer and treated using reverse crosslinking buffer at 55 °C for 45 min, then incubated at 65 °C for 45 min followed by incubation on ice for 5 min. Thereafter, the uncrosslinked DNA was purified, washed and eluted. The Q-PCR analysis were performed to detect the occupancy of NKRF on the negative regulatory elements (NRE) of *hiNOS* and *IFN-β* promoter. The primers used are listed in Supplementary Data 4.

### TdT-mediated dUTP nick-end labeling (Tunel) assays
Tunel assay of paraffin-embedded sections was performed using a DAB (SA-HRP) Tunel Cell Apoptosis Detection Kit (Servicebio Technology, Wuhan, China) according to the manufacturer's directions. Briefly, the

sections were treated with proteinase K, permeabilized with 0.1–0.2% Triton X-100 and incubated with TdT reaction cocktails for 60 min at 37 °C. The HRP-Streptavidin staining solution were added for 1 h at room temperature and hematoxylin was then used to counterstain the nucle. The images were observed to count the number of Tunel positive cells.

### Statistics and reproducibility
The Student's *t* test was used to compare the differences between two independent groups, one-way analysis of variance (ANOVA) and two-way ANOVA were used for comparisons among three or more groups, and the results are presented as the mean values ± S.D. We categorized GCLM level as low or high compared with the median value of IHC score and estimated survival curves using the Kaplan–Meier method (log-rank test) and the estimation of disease-free survival didn't include patients in stage IV. We evaluated independent prognostic factors via univariate and multivariate Cox regression analyses. Chi-square test was used to investigate the correlation between two continuous variables. All statistical tests were two-tailed. All experiments were performed at least three times as independent experiments with similar results, and representative images are shown. The animal numbers were estimated based on previous experience with those models[49,50]. The randomization of allocation was performed by random number generation by computer. Statistical analyses were performed using GraphPad Prism version 8.3.0 (La Jolla, CA, USA) and IBM SPSS Statistics version 21.0 (Armonk, NY, USA).

### Reporting summary
Further information on research design is available in the Nature Portfolio Reporting Summary linked to this article.

## Data availability
Analysis of the expression of candidate genes from the CRISPR–Cas9 screen in colon and rectum adenocarcinoma patients was performed with GEPIA database (http://gepia2.cancer-pku.cn/#analysis). The putative NLS in GCLM was predicted by cNLS Mapper (https://nls-mapper.iab.keio.ac.jp/cgi-bin/NLS_Mapper_form.cgi). The putative phosphorylation sites in GCLM were analyzed using PhosphoSitePlus (https://www.phosphosite.org/proteinAction.action?id=4849775&showAllSites=true). The sequencing raw data related to the CRISPR–Cas9 screen results has been deposited to Gene Expression Omnibus under the accession number GSE254136. The mass spectrometry raw data have been deposited to the ProteomeXchange Consortium via the PRIDE partner repository with the dataset identifier PXD055376. The antibodies and reagents used in this study are listed in Supplementary Data 4. The data generated in the current study are available within the article and its Supplementary materials. Source data are provided with this paper.

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

## Acknowledgements

This project was supported by the National Key R&D Program of China (2023YFC3402100), National Natural Science Foundation of China (82303306, 82173128, 82103643, 82373376, 82370698, 32370698), Guangdong Basic and Applied Basic Research Foundation (2023A1515030257), Postdoctoral Science Foundation of China (BX20220362, 2022M723630), Sanming Project of Medicine in Shenzhen (SZSM202211017), Young Talents Program of Sun Yat-sen University Cancer Center (YTP-SYSUCC-0029), and Hunan Provincial Natural Science Foundation of China (2024JJ4026).

## Author contributions

R.-H.X., H.-Q.J., and J.-F.L. designed the study and composed the paper. J.-F.L., Z.-X.L., and D.-L.C. conducted experiments and analyzed the data. R.-Z.H. and F.C. conducted the revised experiments. T.L., H.-Y.M., H.S., and Y.H. assisted the experiments. J.-F.L., Z.-X.L., K.Y., and K.L. performed the statistical analysis. Z.-L.Z., Z.-B.L., S.-S.L., and S.G. revised the manuscript. R.-H.X. and H.-Q.J. supervised the project. All authors reviewed the manuscript and approved the final version.

## Competing interests

The authors declare no competing interests.

## Additional information

**Peer review information** *Nature Communications* thanks Rumela Chakrabarti, Daqian Xu and the other, anonymous, reviewer(s) for their contribution to the peer review of this.

