## [Peer review file · Nature Communications]

Nucleus-Translocated GCLM Promotes Chemoresistance in Colorectal Cancer through a Moonlighting Function

Corresponding Author: Professor Rui-Hua Xu

Version 0:

Reviewer comments:

Reviewer #1

(Remarks to the Author)

This manuscript by Lin et al., titled "Nucleus-Translocated GCLM Promotes Chemoresistance in Colorectal Cancer through a Moonlighting Function", proposes that glutamate-cysteine ligase modifier (GCLM) promotes resistance to platinum chemotherapy in colorectal cancer (CRC) through a non-enzymatic function. Platinum drugs induce p38 activity which phosphorylates GCLM at threonine 17 (T17). This phosphorylation event is required for augmenting GCLM's interaction with importin $\alpha 5$ and its nuclear accumulation. Nuclear GCLM directly interacts with NF-kappa-B-repressing factor (NKRF) and disrupts its repressive effect on NF-kB signaling. The manuscript presents an enormous amount of biochemical data to delineate the putative mechanism and the proposed mechanism would be of interest to the readers. Additionally, the authors also use a range of in vitro and in vivo models to strength their pre-clinical data. However, there are several concerns which should be addressed before publication.

Major concerns:

1. As the canonical metabolic enzyme, GCLM interacts with GCLC to modify the catalytic efficiency of GCLC. Whether GCLC was included in the CRISPR screening results? It would be helpful to have some (even limited) in vitro experiments demonstrating whether inhibition of GCLC causes the same phenotype as the GCLM inhibition. These experiments would better support the proposed mechanism that GCLM is doing non-enzymatic nuclear functions.
2. This study showed that the expression of nuclear GCLM was elevated in CRC tissues and high nuclear GCLM levels correlates with poor prognosis in patients with CRC. It would be helpful to investigate the expression of nuclear GCLM in other cancer types to show whether the role of nuclear GCLM is special for CRC.
3. Nucleus-Translocated ACSS2 and fumarase also function as metabolic enzymes to produce succinyl-CoA, acetyl-CoA and fumarate, which result in histone acetylation and methylation, respectively (PMID: 28552616, PMID: 26237645). GCLM is the rate-limiting enzyme in the biosynthesis of GSH. Hence, the authors should explore whether GCLM also functions as a metabolic enzyme to produce GSH in the nucleus.
4. NKRF was first found to interact with a specific negative regulatory element at the promoters of NF-kB target genes to repress their transcription via its DNA-binding ability. The author should explore whether GCLM-NKRF interaction affects the combination of NKRF with the promoters of NF-kB target genes.
5. The authors showed that nuclear localization of GCLM in patients with progressive disease. The IHC shows diffuse staining. It would be helpful to explain how the images in Fig 2c, 7b were determined to show GCLM in the nucleus.
6. GCLM was found to display perinuclear focal localization in hepatocytes in previous study (PMID: 17464988). Then, the authors should state the innovative aspects of their paper.

Minor concerns:

1. Line 138: "orthotopic model" is better described as "CDX model" according to the experimental and context.
2. Line 195: it should add "Supplemental Fig. 2G" in "Moreover, C193/194A mutation had little effect on the function of nuclear GCLM (Fig. 2G)".
3. Appropriate spacing should be added between IF and IHC images
4. Arrows or some other indicator should be added to point out the adenomas in HE figures of Fig. 6E.
5. It would be helpful to add a description that importin $\alpha 5$ is encoded by gene KPNA1.

Reviewer #2

(Remarks to the Author)

In this manuscript, Lin et al. present compelling evidence indicating that the loss of glutamate-cysteine ligase modifier subunit (GCLM) heightens sensitivity to platinum-based chemotherapy by promoting the translocation of GCLM to the nucleus, thereby boosting NF κ B activity. They conduct thorough biochemical experiments to investigate the mechanism underlying GCLM's nuclear translocation and its additional functions there. Several key findings are corroborated in xenograft models, patient samples, and a mouse model featuring conditional deletion of GCLM in the intestine. While the findings are intriguing, there are several points that need addressing before publication.

Major Points:

1. Figure 1C demonstrates that knocking out GCLM has the most significant impact on enhancing the sensitivity of HCT116 cells to oxaliplatin. However, there are no corresponding quality control experiments to confirm that knocking down GCLM and other candidate genes reduce the expression of these genes in Figure 1C.
2. It remains unclear whether GCLC, which directly interacts with GCLM to modulate the catalytic efficiency of GCLC in the cytoplasm, was included in the screening. Interfering with GCLC does not appear to affect the chemosensitivity of colorectal cancer to oxaliplatin.
3. The results presented in Figure 3C lack precision; notably, the quantity of GCLM is significantly higher than that of GST, and the input of NKRF is inconsistent. Using antibodies for result presentation would enhance accuracy and clarity.
4. There's a deficiency in validating the necessity of NKRF as a target of GCLM. It's suggested to enhance the detection of chemotherapy sensitivity by knocking down GCLM under conditions where NKRF is deficient.
5. The authors should explore the expression of nuclear GCLM in other cancers or cancer cell lines to validate the specificity of nuclear GCLM as a prognostic factor in CRC.
6. Including a working model to illustrate the methodology used to establish CRC PDX models is recommended.

Minor Points:

1. Consider shortening some lengthy sentences in the manuscript to improve readability.
2. Clarify why experiments concluded when tumors were still small in the PDX model.
3. Correct "the combination GCLM inhibition and oxaliplatin treatment group" to "the combination of GCLM inhibition and oxaliplatin treatment group" in line 142.
4. Amend "in the GCLM-depleted and oxaliplatin treatment groups" to "in the GCLM-depleted, oxaliplatin treatment, and the combination treatment groups" to align with the context in line 146.

Reviewer #3

(Remarks to the Author)

Overall, the manuscript provides a comprehensive overview of the study's findings, detailing experiments conducted to investigate the role of GCLM in chemosensitivity in colorectal cancer (CRC). All sections are well-structured, with clear subheadings delineating different aspects of the study. The language used is technical but understandable, which is suitable for a scientific manuscript. However, some sentences are lengthy and could be broken down for better clarity and readability.

The study employs a variety of experimental techniques, including CRISPR-Cas9 screening, cell viability assays, in vivo models, immunoprecipitation, and molecular assays, which provide a robust foundation for the conclusions drawn. The use of both in vitro and in vivo models strengthens the translational relevance of the findings. However, the manuscript lacks details on certain experimental procedures and controls, which are essential for ensuring the validity and reproducibility of the results.

Based on the findings presented in the manuscript, here are some specific comments that could further enhance the study and address potential limitations:

1. An additional mechanistic experiment to validate the key findings, particularly to the role of GCLM in chemosensitivity and its interaction with NKRF would be preferable. Independent validation using at least in vitro approaches would strengthen the robustness of the conclusions.
2. Mutant forms of GCLM with specific phosphorylation site mutations (T17A) will help to investigate the functional significance of phosphorylation in GCLM nuclear translocation and chemoresistance. Comparing the effects of wild-type and mutant GCLM on cellular phenotypes could provide insights into the mechanistic basis of GCLM-mediated chemoresistance.
3. In vivo pharmacological studies using P38 inhibitors targeting GCLM phosphorylation to assess their efficacy in sensitizing CRC tumors to platinum-based chemotherapy can also be a possible explanation. Evaluate the combination therapy regimen in mouse xenograft models to validate its therapeutic potential in a preclinical setting.
4. The manuscript generalizes findings from in vitro and animal models to clinical settings without sufficient validation in human cohorts. While preclinical models provide valuable insights, translating these findings to human patients requires rigorous clinical validation and may not always be straightforward. Hence, it is needed to perform correlation analyses using clinical data from CRC patient cohorts to validate the prognostic significance of nuclear GCLM expression and T17 phosphorylation. Investigation whether these biomarkers can predict patient responses to platinum-based chemotherapy and overall survival outcomes. Additionally, if available, functional studies using patient-derived samples, such as tumor organoids or patient-derived xenografts, could provide more clinically relevant insights into the efficacy of therapeutic interventions. This would help bridge the gap between preclinical research findings and clinical applications.
5. What is the functional significance of importin α 5 in mediating GCLM nuclear translocation? There is a need for characterization of the interaction between GCLM and importin α 5 in more detail to elucidate the specific molecular determinants involved.
6. The manuscript describes the localization of GCLM in the nucleus in tissues from patients with lesser benefit from

chemotherapy. However, the nuclear localization of GCLM observed in IHC staining may not necessarily indicate its functional activity or relevance to chemoresistance. This interpretation should be carefully done and supported by additional functional studies.

7. In vivo models, i.e., mouse xenografts or spontaneous CRC models, should be carefully validated to ensure their relevance and reliability in recapitulating human disease phenotypes. This could include histopathological characterization of tumor tissues, assessment of tumor growth kinetics, and comparison of treatment responses with clinical standards of care.

8. When generating mutant constructs, i.e., the GCLM-T17A or GCLM-T17E mutants, it's important to perform additional characterization experiments to confirm the expected changes in protein function or localization. This could include validation of protein expression levels, subcellular localization, and functional assays to assess the impact of mutations on protein activity.

9. It's crucial to include negative controls in immunoprecipitation and pull-down assays to ensure the specificity of protein-protein interactions. Negative controls could include non-specific immunoglobulin G (IgG) or unrelated proteins as negative binding controls.

10. DAPI staining is commonly used to assess nuclear morphology, but it is not a specific marker for cell viability. To accurately measure cell viability, complementary assays such as MTT, MTS, or ATP assays should be performed. These assays directly measure metabolic activity or ATP levels, providing a more reliable indicator of cell viability.

11. Hematoxylin and eosin (H&E) staining is a standard method for histological analysis, but it may not provide sufficient specificity for identifying fibrosis. Additional staining techniques such as Masson's trichrome or Sirius Red staining, which specifically target collagen fibers, would enable more accurate quantification and characterization of fibrosis in tumor tissues.

Version 1:

Reviewer comments:

Reviewer #1

(Remarks to the Author)

Reviewer #2

(Remarks to the Author)

The authors have addressed all of my concerns—congratulations!

Reviewer #3

(Remarks to the Author)

Authors have successfully responded to all comments of the reviewer.

Manuscript ID: NCOMMS-24-17740

Title: Nucleus-Translocated GCLM Promotes Chemoresistance in Colorectal Cancer through a Moonlighting Function

Detailed point-by-point responses to the reviewers' comments:

Note to reviewers: We thank the reviewers for taking the time to review our manuscript and providing insightful comments to further improve our manuscript. In this letter, we provide detailed point-by-point responses to all the comments raised by the reviewers. We present all the new data in response letter figures in this letter and refer to the corresponding figures and text in our revised manuscript to facilitate review of our response letter and manuscript by the reviewers. We have also marked all corresponding changes in our revised manuscript by in colored text.

Reviewer #1 - Cancer metabolism (Remarks to the Author):

This manuscript by Lin et al., titled “Nucleus-Translocated GCLM Promotes Chemoresistance in Colorectal Cancer through a Moonlighting Function”, proposes that glutamate-cysteine ligase modifier (GCLM) promotes resistance to platinum chemotherapy in colorectal cancer (CRC) through a non-enzymatic function. Platinum drugs induce p38 activity which phosphorylates GCLM at threonine 17 (T17). This phosphorylation event is required for augmenting GCLM’s interaction with importin $\alpha 5$ and its nuclear accumulation. Nuclear GCLM directly interacts with NF-kappa-B-repressing factor (NKRF) and disrupts its repressive effect on NF-kB signaling. The manuscript presents an enormous amount of biochemical data to delineate the putative mechanism and the proposed mechanism would be of interest to the readers. Additionally, the authors also use a range of in vitro and in vivo models to strength their pre-clinical data. However, there are several concerns which should be addressed before publication.

Response: Thank you very much for your review of our manuscript. We also thank

you for the positive comments and the following constructive suggestions listed below, which were very helpful in strengthening our manuscript.

Major concerns:

1.As the canonical metabolic enzyme, GCLM interacts with GCLC to modify the catalytic efficiency of GCLC. Whether GCLC was included in the CRISPR screening results? It would be helpful to have some (even limited) in vitro experiments demonstrating whether inhibition of GCLC causes the same phenotype as the GCLM inhibition. These experiments would better support the proposed mechanism that GCLM is doing non-enzymatic nuclear functions.

Response: Thank you for providing this valuable suggestion. We reanalyzed the CRISPR–Cas9 screening results. Compared with the control group, we found that GCLC also decreased in the three oxaliplatin treatment groups with latter ranking, and the log₂ fold changes (LFCs) were -2.5677 , -3.4172 and -3.2116 , respectively (revised Supplementary Data 1). As suggested, we knocked down GCLC and GCLM to evaluate the function of GCLC and GCLM in the behaviors of CRC cells (**response Fig. 1A**; Supplementary Fig. 1b in revised manuscript). The results showed that either GCLC or GCLM inhibition could reduce the cell viability and increase the apoptotic rate in HCT116 and DLD1 cells under the treatment of oxaliplatin, and GCLM inhibition had a better suppressive effect (**response Fig. 1B-D**; Supplementary Fig. 1c in revised manuscript), which was consistent with our CRISPR–Cas9 screening results and implied that GCLM may have a moonlighting function.

Figure legend 1. (A) Q-PCR analysis of GCLM or GCLC expression after knocking down GCLM or GCLC in HCT116 and DLD1 cells. (B-D) Cell viability (B) and apoptosis rate (C, D) analysis of HCT116 or DLD1 cells after knocking down GCLM or GCLC with oxaliplatin treatment.

2. This study showed that the expression of nuclear GCLM was elevated in CRC tissues and high nuclear GCLM levels correlates with poor prognosis in patients with CRC. It would be helpful to investigate the expression of nuclear GCLM in other cancer types to show whether the role of nuclear GCLM is special for CRC.

Response: Thanks for your suggestions. We have added gastric carcinoma (GC) and esophageal squamous cell carcinoma (ESCC) patient cohorts from Sun Yat-sen University Cancer Center (SYSUCC) to investigate the role of nuclear GCLM in patient prognosis. Similarly, we categorized the nuclear GCLM levels as low or high in comparison with the median value, and the Kaplan–Meier survival analysis indicated that a high level of nuclear GCLM expression was associated with a poor prognosis for patients with GC and ESCC, although no notable statistical significance was observed in patients with ESCC, probably due to insufficient sample size (**response Fig. 2A-C; Fig. 7e and Supplementary Fig. 7a, b in revised manuscript**). Collectively, these data further indicate that nuclear GCLM has a widespread prognostic indicator potential.

Figure legend 2. (A) Representative IHC staining images of nuclear GCLM expression in (nGCLM, just focus on the GCLM staining in nucleus, as shown by blue arrowheads) in GC and ESCC tissues. (B,C) Overall survival assays of patients with GC or ESCC based on nGCLM expression.

3. Nucleus-Translocated ACSS2 and fumarase also function as metabolic enzymes to produce succinyl-CoA, acetyl-CoA and fumarate, which result in histone acetylation and methylation, respectively (PMID: 28552616, PMID: 26237645). GCLM is the rate-limiting enzyme in the biosynthesis of GSH. Hence, the authors should explore whether GCLM also functions as a metabolic enzyme to produce GSH in the nucleus.

Responses: We thank you for providing this valuable suggestion. As suggested, we selectively increased the GCLM level in the nucleus by expressing shRNA-resistant GCLM (rGCLM) fused to a nuclear localization signal (rGCLM-NLS) in endogenous GCLM knockdown HCT116 cells, leaving the level of cytoplasmic GCLM unchanged (**response Fig. 3A**; Supplementary Fig. 2f in revised manuscript). The results showed that overexpressing nuclear GCLM (rGCLM-NLS) had a negligible effect on the level of GSH ($P = 0.3633$, **response Fig. 3B**; Supplementary Fig. 2g in revised manuscript), suggesting that GCLM does not function as a metabolic enzyme to produce GSH in the nucleus. We have supplemented above results and descriptions to the revised manuscript (revised page 9, lines 190-191).

Figure legend 3. (A) Representative IF staining images of the localization of GCLM in HCT116 cells overexpressing nuclear GCLM (+NLS). (B) The GSH level analysis of GCLM knockdown HCT116 cells overexpressing control, rGCLM WT or nuclear GCLM (rGCLM NLS).

4. NKRF was first found to interact with a specific negative regulatory element at the promoters of NF- κ B target genes to repress their transcription via its DNA-binding ability. The author should explore whether GCLM-NKRF interaction affects the

combination of NKRF with the promoters of NF- κ B target genes.

Responses: Thanks for pointing out this insightful issue. We designed specific primers for the negative regulatory elements (NREs) in the promoters of NF- κ B target genes (iNOS and IFN- β)¹. ChIP-PCR assay showed that GCLM inhibition had no significant effect on the interaction between NKRF and NRE on the promoter of iNOS or IFN- β (**response Fig. 4**; Supplementary Fig. 3j in revised manuscript). This suggest that GCLM-NKRF interaction does not affect the DNA-binding ability of NKRF. We have included these data in our revised manuscript (revised page11, lines 235-237).

Figure legend 4. Q-PCR analysis after ChIP analysis showing the occupancy of NKRF on the negative regulatory elements (NRE) of iNOS, IFN- β promoter in HCT116 cells.

5.The authors showed that nuclear localization of GCLM in patients with progressive disease. The IHC shows diffuse staining. It would be helpful to explain how the images in Fig 2c, 7b were determined to show GCLM in the nucleus.

Response: We apologize for the confusion. Since an antibody that is specific for nuclear GCLM is not available, we calculated the expression of nuclear GCLM by focusing on the GCLM staining in nucleus, as indicated by the blue arrowheads. We have provided clearer descriptions in our revised figure legend (revised page 44, line 1106). We hope that you will agree with our interpretation.

6.GCLM was found to display perinuclear focal localization in hepatocytes in previous study (PMID: 17464988). Then, the authors should state the innovative

aspects of their paper.

Response: We apologize for not discussing this issue enough and making you confuse. The previous study only showed that GCLM displayed perinuclear focal localization in hepatocytes, and the specific molecular mechanism of GCLM nuclear translocation and the functions of GCLM in the nucleus have not been elucidated². In our work, we further revealed a moonlighting function of GCLM in nucleus through the NKRF–NF- κ B axis, which contributes to CRC chemoresistance and progression. In addition, platinum drug treatment promotes GCLM nuclear localization via GCLM binding to importin $\alpha 5$ in a P38 MAPK-mediated phosphorylation-dependent manner. We have supplemented above discussion in our revised manuscript (revised page 19, lines 427-428).

Minor concerns:

1. Line 138: “orthotopic model” is better described as “CDX model” according to the experimental and context.

Response: We sorry for this inappropriate description. We have corrected “orthotopic model” to “CDX model” in revised page 7, line 139.

2. Line 195: it should add “Supplemental Fig. 2G” in “Moreover, C193/194A mutation had little effect on the function of nuclear GCLM (Fig. 2G)”.

Response: We apologize for missing the supplementary figure. We have added this supplementary figure to page 9, line 197 in the revised manuscript.

3. Appropriate spacing should be added between IF and IHC images

Response: Thanks for your suggestions. We have adjusted the spacing between the images of IF and IHC staining in revised figures.

4. Arrows or some other indicator should be added to point out the adenomas in H&E figures of Fig. 6E.

Response: Thanks for your suggestions. We have added blue arrows to indicate the adenomas in the lower power H&E images for readers to view (**response Fig. 5; Fig. 7c** in revised manuscript).

Figure legend 5. Representative images of H&E staining of spontaneous mouse CRC model. The blue arrows indicate the adenomas.

5. It would be helpful to add a description that importin $\alpha 5$ is encoded by gene KPNA1.

Response: We apologize for not providing this information. We have added the description “importin $\alpha 5$ also known as KPNA1” for readers to understand in revised page 13, lines 281.

References

1. Feng X, *et al.* Identification of a negative response element in the human inducible nitric-oxide synthase (hiNOS) promoter: The role of NF-kappa B-repressing factor (NRF) in basal repression of the hiNOS gene. *Proc Natl Acad Sci U S A* **99**, 14212-14217 (2002).
2. Chen Y, *et al.* Hepatocyte-specific Gclc deletion leads to rapid onset of steatosis with mitochondrial injury and liver failure. *Hepatology* **45**, 1118-1128 (2007).

Reviewer #2 - Colorectal cancer, metabolism & resistance (Remarks to the Author):

In this manuscript, Lin et al. present compelling evidence indicating that the loss of glutamate-cysteine ligase modifier subunit (GCLM) heightens sensitivity to platinum-based chemotherapy by promoting the translocation of GCLM to the

nucleus, thereby boosting NFKB activity. They conduct thorough biochemical experiments to investigate the mechanism underlying GCLM's nuclear translocation and its additional functions there. Several key findings are corroborated in xenograft models, patient samples, and a mouse model featuring conditional deletion of GCLM in the intestine. While the findings are intriguing, there are several points that need addressing before publication.

Response: We appreciate the positive and insightful comments from this reviewer and hope that our revision has largely addressed the critiques from this reviewer.

Major Points:

1. Figure 1C demonstrates that knocking out GCLM has the most significant impact on enhancing the sensitivity of HCT116 cells to oxaliplatin. However, there are no corresponding quality control experiments to confirm that knocking down GCLM and other candidate genes reduce the expression of these genes in Figure 1C.

Response: Thank you for noting this issue. We have complemented the results of the Q-PCR assay to verify the knockout efficiency of the 10 candidate genes, and the results showed similar knockout efficiencies (**response Fig. 1; Supplementary Fig. 1a in revised manuscript**).

Figure legend 1. Q-PCR analysis of the candidate gene expression after knocking out CYP3A5, APEH, ASS1, ALG1, GGH, GFPT1, GSTO2, AGMAT, SHMT2 or GCLM in HCT116 cells.

2. It remains unclear whether GCLC, which directly interacts with GCLM to modulate

the catalytic efficiency of GCLC in the cytoplasm, was included in the screening. Interfering with GCLC does not appear to affect the chemosensitivity of colorectal cancer to oxaliplatin.

Responses: We thank you for raising this crucial question, and we sincerely apologize for the inadequate experiment. We reanalyzed the CRISPR–Cas9 screening results. We found that GCLC also decreased in the three oxaliplatin treatment groups compared with the control group with latter ranking, and the log2 fold changes (LFCs) was -2.5677 , -3.4172 and -3.2116 , respectively (revised Supplementary Data 1). In addition, we knocked down GCLC to evaluate its effect on chemosensitivity (**response Fig. 2A**; Supplementary Fig. 1b in revised manuscript). The results showed that GCLC inhibition could also enhance the chemosensitivity of CRC to oxaliplatin by reducing the cell viability and increasing the apoptotic rate in HCT116 and DLD1 cells, but GCLM inhibition exerted a better enhancement effect (**response Fig. 2B-D**; Supplementary Fig. 1c in revised manuscript). These results also implied that GCLM may have a moonlighting function in addition to its metabolic enzyme activity. We have included related results and descriptions in the revised manuscript (revised page 6, lines 125-127.)

Figure legend 2. (A) Q-PCR analysis of GCLM or GCLC expression after knocking down GCLM or GCLC in HCT116 and DLD1 cells. (B-D) Cell viability (B) and apoptosis rate (C, D)

analysis of HCT116 or DLD1 cells after knocking down GCLM or GCLC with oxaliplatin treatment.

3. The results presented in Figure 3C lack precision; notably, the quantity of GCLM is significantly higher than that of GST, and the input of NKRF is inconsistent. Using antibodies for result presentation would enhance accuracy and clarity.

Responses: We apologize for the inappropriate experiment. As suggested, we repeated the GST pull-down experiment with the corresponding antibodies (**response Fig. 3A**; Fig. 3c in revised manuscript). The results showed that GCLM indeed directly interacts with NKRF, which is consistent with our conclusions.

Figure legend 3. GST pull-down analysis detecting the level of purified His-NKRF interacted with purified GST-GCLM or GST

4. There's a deficiency in validating the necessity of NKRF as a target of GCLM. It's suggested to enhance the detection of chemotherapy sensitivity by knocking down GCLM under conditions where NKRF is deficient.

Responses: We thank you for providing this constructive suggestion. Our original data revealed that nuclear GCLM interacts with NKRF to relieve the repressive effect of NKRF on NF- κ B/p65 activity and contributes to chemoresistance. As suggested, we overexpressed NKRF when nuclear GCLM (rGCLM-NLS) was overexpressed. The results showed that the increased NF- κ B/p65 activity induced by re-expressing nuclear rGCLM could be reversed by overexpressing NKRF (**response Fig. 4A**; Fig. 3k in revised manuscript). Consistently, the results of the cellular phenotype assays

revealed that silencing NKRF could partially reverse the decreased ATP level, cell viability and increased apoptosis rate in HCT116 cells induced by GCLM inhibition (response Fig. 4B-D; Supplementary Fig. 3g-i in revised manuscript). Above results further validated the necessity of NKRF as a target of nuclear GCLM. We have included related results and descriptions in revised manuscript (revised page 11, lines 226-228, 247-249).

Figure legend 4. (A) HCT116 or 293T cells were depleted GCLM and re-expressed nuclear GCLM (NLS) with control or NKRF inhibition in the presence of oxaliplatin. The transcriptional activity of NF-κB/p65 was detected by Dual-luciferase assay. (B-D) GCLM was depleted in HCT116 cells with or without importin α5 inhibition under oxaliplatin treatment. ATP level (B), cell viability (C) and apoptotic cell (D) were detected.

5. The authors should explore the expression of nuclear GCLM in other cancers or cancer cell lines to validate the specificity of nuclear GCLM as a prognostic factor in CRC.

Response: Thanks for your precious suggestions. We have added gastric carcinoma (GC) and esophageal squamous cell carcinoma (ESCC) patient cohorts from SYSUCC to investigate the role of nuclear GCLM in patients prognosis. Similarly, we categorized the nuclear GCLM levels as low or high in comparison compared with the

median value, and Kaplan–Meier survival analysis indicated that a high level of nuclear GCLM expression was associated with a poor prognosis in patients with GC and ESCC, although no notable statistical significance was observed in patients with ESCC, probably due to insufficient sample size (**response Fig. 5A-C**; Fig. 7e and Supplementary Fig. 7a, b in revised manuscript). Collectively, these data further indicate that nuclear GCLM has a widespread prognostic indicator potential.

Figure legend 5. (A) Representative IHC staining images of nuclear GCLM expression in (nGCLM, just focus on the GCLM staining in nucleus, as shown by blue arrowheads in GC and ESCC tissues). (B,C) Overall survival assays of patients with GC or ESCC based on nGCLM expression.

6. Including a working model to illustrate the methodology used to establish CRC PDX models is recommended.

Response: We thank you for this important suggestion. We have supplemented a working model to show the process of establishing PDX mouse model as previously reported¹ (**response Fig. 6**; Supplementary Fig. 1g in revised manuscript), and the detailed process was described in the revised Methods section.

Figure legend 6. Illustration of the methodology used to establish CRC PDX models.

Minor Points:

1. Consider shortening some lengthy sentences in the manuscript to improve readability.

Response: Thank you for pointing out this issue. We have shortened the lengthy sentences and carefully improved the writing in the revised manuscript.

2. Clarify why experiments concluded when tumors were still small in the PDX model.

Responses: We thank you for pointing out this issue. The PDX models we established had a slower tumor growth rate than the CDX model did. We ended the experiment after 7 intraperitoneal (i.p.) injections of PBS or oxaliplatin similar to the CDX model to reduce the excessive side effects of oxaliplatin. We hope that you will concur with our interpretation.

3. Correct "the combination GCLM inhibition and oxaliplatin treatment group" to "the combination of GCLM inhibition and oxaliplatin treatment group" in line 142.

Response: We apologize for the grammatical issues. We have corrected "the combination GCLM inhibition and oxaliplatin treatment group" to "the combination of GCLM inhibition and oxaliplatin treatment group" in manuscript (revised page 7, lines 143), and we have carefully checked the entire manuscript.

4. Amend "in the GCLM-depleted and oxaliplatin treatment groups" to "in the GCLM-depleted, oxaliplatin treatment, and the combination treatment groups" to align with the context in line 146.

Response: Thank you for pointing out this issue. We have corrected "in the GCLM-depleted and oxaliplatin treatment groups" to "in the GCLM-depleted, oxaliplatin treatment, and the combination treatment groups" to align with the context in revised manuscript (revised page 7, lines 147).

References

1. Wang Y, *et al.* Inhibition of fatty acid catabolism augments the efficacy of oxaliplatin-based chemotherapy in gastrointestinal cancers. *Cancer Lett* **473**, 74-89 (2020).

Reviewer #3 - NF-KB, chemoresistance (Remarks to the Author):

Overall, the manuscript provides a comprehensive overview of the study's findings, detailing experiments conducted to investigate the role of GCLM in chemosensitivity in colorectal cancer (CRC). All sections are well-structured, with clear subheadings delineating different aspects of the study. The language used is technical but understandable, which is suitable for a scientific manuscript. However, some sentences are lengthy and could be broken down for better clarity and readability. The study employs a variety of experimental techniques, including CRISPR-Cas9 screening, cell viability assays, in vivo models, immunoprecipitation, and molecular assays, which provide a robust foundation for the conclusions drawn. The use of both in vitro and in vivo models strengthens the translational relevance of the findings. However, the manuscript lacks details on certain experimental procedures and controls, which are essential for ensuring the validity and reproducibility of the results. Based on the findings presented in the manuscript, here are some specific comments that could further enhance the study and address potential limitations:

Responses: We thank you for appreciating our study and providing valuable comments to enhance the quality of our manuscript. We have shortened the lengthy sentences and carefully improved the writing in the revised manuscript for better clarity and readability. In addition, we have modified the listed issues based on your suggestion, including the lack of details on certain experimental procedures and controls. We hope that our revision has largely addressed your critiques.

1. An additional mechanistic experiment to validate the key findings, particularly to the role of GCLM in chemosensitivity and its interaction with NKRF would be preferable. Independent validation using at least in vitro approaches would strengthen the robustness of the conclusions.

Response: Thank you very much for your constructive suggestions. We repeated the GST pull-down experiment using a more appropriate method, and the results showed

that GCLM indeed directly interacts with NKRF (**response Fig. 1A**; Fig. 3c in revised manuscript). In addition, our original data showed that nuclear GCLM interacts with NKRF to relieve the repressive effect of NKRF on NF- κ B/p65 activity and contributes to chemoresistance. As suggested, we overexpressed NKRF when shRNA-resistant nuclear GCLM (rGCLM-NLS) was overexpressed. The results showed that the increased NF- κ B/p65 activity induced by re-expressing nuclear GCLM could be reversed by overexpressing NKRF (**response Fig. 1B**; Fig. 3k in revised manuscript). Consistently, the results of the cellular phenotype assays revealed that silencing NKRF partially reversed the decreases in ATP levels and cell viability and the increase in the apoptosis rate in HCT116 cells induced by GCLM inhibition (**response Fig. 1C-E**; Supplementary Fig. 3g-i in revised manuscript). The above results further validated the necessity of NKRF as a target of nuclear GCLM to facilitate chemoresistance in CRC. We have included related results and descriptions in the revised manuscript (revised page 11, lines 226-228, 247-249).

Figure legend 1. (A) GST pull-down analysis detecting the level of purified His-NKRF interacted with purified GST-GCLM or GST. (B) HCT116 or 293T cells were depleted GCLM and re-expressed nuclear GCLM (NLS) with control or NKRF inhibition in the presence of oxaliplatin. The transcriptional activity of NF- κ B/p65 was detected by Dual-luciferase assay. (C-E) GCLM was depleted in HCT116 cells with or without importin α 5 inhibition under oxaliplatin treatment.

ATP level (C), cell viability (D) and apoptotic cells (E) were detected.

2. Mutant forms of GCLM with specific phosphorylation site mutations (T17A) will help to investigate the functional significance of phosphorylation in GCLM nuclear translocation and chemoresistance. Comparing the effects of wild-type and mutant GCLM on cellular phenotypes could provide insights into the mechanistic basis of GCLM-mediated chemoresistance.

Response: We thank you for providing this valuable suggestion. We have depleted endogenous GCLM and re-expressed shRNA-resistant rGCLM WT, T17A and T17E in HCT116 cells. Compared with re-expression of rGCLM WT, re-expression of rGCLM T17A but not re-expression of rGCLM T17E decreased cell viability, ATP level and increased the apoptosis rate, and these effects were more significant following oxaliplatin treatment (**response Fig. 2A-C**; Fig. 5k, l and Supplementary Fig. 5h-j in revised manuscript). These cellular phenotypes further elucidated that the phosphorylation of GCLM at T17 is crucial for the function of nuclear GCLM. These data have been included in the revised manuscript (revised page 14, lines 341-344).

Figure legend 2. (A-C) cell viability (A), ATP level (B) and apoptotic cells (C) were detected in GCLM knockdown HCT116 cells overexpressing rGCLM WT or T17A, T17E mutants with or without oxaliplatin treatment (40 μ M, 24 h).

3. In vivo pharmacological studies using P38 inhibitors targeting GCLM phosphorylation to assess their efficacy in sensitizing CRC tumors to platinum-based chemotherapy can also be a possible explanation. Evaluate the combination therapy regimen in mouse xenograft models to validate its therapeutic potential in a preclinical setting.

Response: We agree with you on this insightful point and have performed additional

experiments. To better emulate the physical tumor microenvironment, we established two patient-derived xenograft (PDX) tumor models to validate the efficacy of a P38 inhibitor (SB203580) on platinum-based chemosensitivity (FOLFOX regimen: oxaliplatin 5 mg/kg, 5-fluorouracil 25 mg/kg)¹. The results showed that P38 inhibitor treatment decreased the tumor volume and weight, and the combination group of the P38 inhibitor and the FOLFOX regime treatment resulted in the greatest reductions in volume and weight in both the PDX #1 and PDX #2 models (response Fig. 3A, B; Fig. 6g, h and Supplementary Fig. 6i in revised manuscript). These findings further validate that targeting GCLM phosphorylation is a potential therapeutic strategy for improving the antitumor efficacy of platinum-based chemotherapy in CRC. We have included these experiments and descriptions in our revised manuscript (revised page 17, lines 375-381).

Figure legend 3. (A-B) Photographs, statistical analysis of the tumor volumes and weights in the PDX #1 (A) and PDX #2 (B) models, followed by intraperitoneal injection of control or P38 inhibitor (P38i, SB203580, 5 mg/kg) and FOLFOX (oxaliplatin 5 mg/kg, 5-fluorouracil 25 mg/kg).

4. The manuscript generalizes findings from in vitro and animal models to clinical settings without sufficient validation in human cohorts. While preclinical models

provide valuable insights, translating these findings to human patients requires rigorous clinical validation and may not always be straightforward. Hence, it is needed to perform correlation analyses using clinical data from CRC patient cohorts to validate the prognostic significance of nuclear GCLM expression and T17 phosphorylation. Investigation whether these biomarkers can predict patient responses to platinum-based chemotherapy and overall survival outcomes.

Additionally, if available, functional studies using patient-derived samples, such as tumor organoids or patient-derived xenografts, could provide more clinically relevant insights into the efficacy of therapeutic interventions. This would help bridge the gap between preclinical research findings and clinical applications.

Responses: We thank you for raising this valuable suggestion, and we sincerely apologize for the unclear description of our experiment. We have used CRC patient cohorts from Sun Yat-sen University Cancer Center (SYSUCC) to investigate the clinical implications of nuclear GCLM and T17-phosphorylated GCLM. The results had showed that the level of GCLM in the nucleus and T17-phosphorylated GCLM were notably increased in primary CRC tissues compared with adjacent normal tissues (**response Fig. 4A**; Fig. 7a, b in revised manuscript). The Kaplan–Meier survival analysis strikingly indicated that patients with high nuclear GCLM levels had unfavorable overall survival, as did patients with high levels of T17-phosphorylated GCLM (**response Fig. 4B**; Fig. 7c, d in revised manuscript). In addition, we further added CRC patient cohorts to validate the clinical correlation between nuclear/T17-phosphorylated GCLM expression and the response to oxaliplatin-based chemotherapy (FOLFOX or XELOX regimens). The results showed that patients with high levels of nuclear GCLM or T17-phosphorylated GCLM staining exhibit a poorer benefit from standard chemotherapy (**response Fig. 4C**; Fig. 7f in revised manuscript). Collectively, these data further indicate that nuclear GCLM and T17-phosphorylated GCLM are promising prognostic indicators that can predict patient responses to platinum-based chemotherapy and overall survival outcomes. (revised page 18, lines 401-405).

As shown in response Fig. 3, we have established two patient-derived xenograft (PDX) tumor models, and added a P38 inhibitor that targets the phosphorylation of GCLM and subsequent nuclear translocation to validate the efficacy of P38 inhibitors on platinum-based chemosensitivity. The results showed that targeting GCLM phosphorylation is a potential therapeutic strategy for improving the antitumor efficacy of platinum-based chemotherapy in CRC.

Figure legend 4. (A) Representative IHC staining images and IHC staining scores of nuclear GCLM expression (nGCLM, just focus on the GCLM staining in nucleus, as shown by blue arrowheads) and phospho-T17-GCLM expression in paired primary CRC tumor tissues (T) and adjacent normal tissues (N). (B) Overall survival assays of patients with CRC based on nuclear GCLM and pT17-GCLM expression. (C) The correlation between nuclear GCLM and pT17-GCLM expression and the response of patients with CRC to FOLFOX or XELOX chemotherapy (PD, progressive disease; SD, stable disease; PR, partial response; CR, complete response).

5. What is the functional significance of importin $\alpha 5$ in mediating GCLM nuclear translocation? There is a need for characterization of the interaction between GCLM and importin $\alpha 5$ in more detail to elucidate the specific molecular determinants involved.

Response: We appreciate these insightful suggestions. As suggested, we supplemented GST pull-down and immunofluorescence assays to verify the interaction of GCLM with importin $\alpha 5$. Immunofluorescence assays revealed

colocalization between GCLM and importin $\alpha 5$ in HCT116 cells, but a GST pull-down assay did not reveal a direct interaction between GCLM and importin $\alpha 5$ (response Fig. 5A, B; Fig. 4h in revised manuscript). This finding consistent with our subsequent results that the GCLM-importin $\alpha 5$ interaction depends on the phosphorylation of GCLM at T17. Consistently, silencing importin $\alpha 5$ notably reduced the nuclear accumulation of GCLM induced by oxaliplatin treatment in HCT116 cells (response Fig. 5C; Fig. 4k in revised manuscript). In addition, we inhibited the expression of importin $\alpha 5$ when rGCLM was overexpressed. The results of the cellular phenotype assays revealed that inhibiting importin $\alpha 5$ obviously reversed the increases in NF- κ B/p65 activity, cell viability, and ATP levels and the reduced apoptosis rate induced by rGCLM overexpression in HCT116 cells treated with oxaliplatin (response Fig. 5D-F; Fig. 4l, m and Supplementary Fig. 4k, l in revised manuscript). Above results further validated the indispensable role of importin $\alpha 5$ in mediating GCLM nuclear translocation to facilitate chemoresistance in CRC. We have included related results and descriptions in the revised manuscript (revised page 13, lines 282-284; 288-291).

Figure legend 5. (A) GST pull-down analysis detecting the level of purified His-importin $\alpha 5$ interacted with purified GST-GCLM or GST. (B) Co-localization of endogenous GCLM and importin $\alpha 5$ in HCT116 cells was detected by IF. Red arrowheads showed the co-localization. (C)

IB detection of nuclear and total GCLM expression in HCT116 cells with control or importin $\alpha 5$ silencing in presence or absence of oxaliplatin treatment. (D-F) GCLM knockdown HCT116 cells overexpressed vector or rGCLM WT with control or importin $\alpha 5$ silencing under oxaliplatin treatment. The transcriptional activity of NF- κ B/p65 (D), ATP level (E), cell viability and apoptotic cells (F) were detected.

6. The manuscript describes the localization of GCLM in the nucleus in tissues from patients with lesser benefit from chemotherapy. However, the nuclear localization of GCLM observed in IHC staining may not necessarily indicate its functional activity or relevance to chemoresistance. This interpretation should be carefully done and supported by additional functional studies.

Response: Thank you for pointing out this important issue. To test whether nuclear GCLM contributes to resistance to platinum-based chemotherapy in CRC cells, we have selectively increased the GCLM level in the nucleus by expressing rGCLM fused to a nuclear localization signal (rGCLM-NLS) (Supplementary Fig. 2f in revised manuscript). The results showed that the overexpression of nuclear GCLM increased the cell viability and decreased the apoptosis rate, as well as in the treatment with oxaliplatin or cisplatin (**response Fig. 6A-C**; Fig. 2g, h and Supplementary Fig. 2i in revised manuscript). Consistently, upon of oxaliplatin treatment, nuclear GCLM overexpression clearly weakened the therapeutic efficiency of chemotherapy compared with that in the control group in vivo (**response Fig. 6D-F**; Fig. 2i in revised manuscript). Above results illustrated that the nuclear localization of GCLM contributes to chemoresistance. These descriptions have been included in the revised manuscript.

Figure legend 6. (A-C) Cell viability (A) and apoptotic cells (B,C) of endogenous GCLM-knockdown HCT116 cells, which was overexpressed control, nuclear GCLM WT (rGCLM NLS) or C193/194A mutant (rGCLM NLS-CA) with PBS, oxaliplatin or cisplatin treatment. (D-F) Photographs and statistical analysis of CDX tumor volumes and weights after implantation of endogenous GCLM-knockdown HCT116 cells, which overexpressed control or nuclear GCLM (rGCLM NLS), followed by intraperitoneal injection of PBS or oxaliplatin (5 mg/kg).

7. In vivo models, i.e., mouse xenografts or spontaneous CRC models, should be carefully validated to ensure their relevance and reliability in recapitulating human disease phenotypes. This could include histopathological characterization of tumor tissues, assessment of tumor growth kinetics, and comparison of treatment responses with clinical standards of care.

Response: We thank you for the valuable suggestion. As suggested, we performed hematoxylin and eosin (H&E) staining to validate the histopathological characteristics of the PDX and spontaneous CRC models, which showed similar histopathological characteristics to those of CRC tissues, especially the PDX model (**response Fig. 7A**; Fig. 1i, 6c in revised manuscript). To validate the treatment responses with clinical standards of care, we added PDX models to confirm the effect of P38 inhibitor (targeting on the phosphorylation of GCLM) on the chemosensitivity to the FOLFOX regimen. The tumor volume was measured periodically to assess the tumor growth kinetics, and the results showed that the P38 inhibitor significantly increased the chemosensitivity of CRC to the FOLFOX regimen (**response Fig. 7B**; Fig. 6g, h in revised manuscript).

Figure legend 7. (A) Representative images of H&E, IHC staining of Ki67 in PDX #1/2 models or spontaneous mouse tumour-based paraffin-embedded subcutaneous tumor sections. (B) Statistical analysis of the tumor volumes and weights in the PDX #1 and PDX #2 models, followed by intraperitoneal injection of control or P38 MAPK inhibitor (P38i, SB203580, 5 mg/kg) and FOLFOX (oxaliplatin 5 mg/kg, 5-fluorouracil 25 mg/kg).

8. When generating mutant constructs, i.e., the GCLM-T17A or GCLM-T17E mutants, it's important to perform additional characterization experiments to confirm the expected changes in protein function or localization. This could include validation of protein expression levels, subcellular localization, and functional assays to assess the

impact of mutations on protein activity.

Response: Thank you very much for your significant suggestions. We have overexpressed shRNA-resistant rGCLM-WT, rGCLM-T17A or rGCLM-T17E mutants in endogenous GCLM-knockdown HCT116 cells. The rGCLM-T17A and rGCLM-T17E mutants had slightly effect on the expression levels of GCLM (**response Fig. 8A**; Supplementary Fig. 5g in revised manuscript). However, overexpression of the rGCLM T17A mutant, but not rGCLM WT or the T17E mutant, significantly diminished the oxaliplatin-induced nuclear accumulation of GCLM (**response Fig. 8B**; Fig. 5e in revised manuscript). Consistently, rGCLM T17A overexpression but not rGCLM T17E overexpression reduced NF- κ B/p65 activity in both the absence or presence of oxaliplatin compared with GCLM WT re-expression (**response Fig. 8C**; Fig. 5j in revised manuscript). As shown in response **response Fig. 2**, the cellular phenotypes showed that compared with re-expression of rGCLM WT, re-expression of rGCLM T17A but not re-expression of rGCLM T17E could decrease cell viability, ATP levels and increased the apoptosis rate. In addition, the in vivo experiments also showed that tumor growth and weights were decreased in the rGCLM T17A re-expression group but not in the rGCLM T17E re-expression group compared with those in the rGCLM WT re-expression group (**response Fig. 8D-F**; Fig. 6e, f and Supplementary Fig. 6g, h in revised manuscript). Collectively, these data show that the phosphorylation of GCLM at T17 plays a crucial role in GCLM nuclear translocation and subsequent chemoresistance in CRC cells.

Figure legend 8. (A) IB detection of GCLM level in control or GCLM knockdown HCT116 cells overexpressing rGCLM WT or T17A, T17E mutants. (B) IB detection of nuclear and total GCLM expression in HCT116 cells overexpressing Flag-tagged rGCLM WT or T17A, T17E mutant. (C) The transcriptional activity of NF-κB/p65 in GCLM knockdown HCT116 cells overexpressing rGCLM WT or T17A, T17E mutants with or without oxaliplatin treatment. (D-F) Photographs and statistical analysis of CDX tumor volumes and weights after implantation of endogenous GCLM-knockdown HCT116 cells, which overexpressed rGCLM WT or T17A, T17E mutants, followed by intraperitoneal injection of PBS or oxaliplatin (5 mg/kg).

9. It's crucial to include negative controls in immunoprecipitation and pull-down assays to ensure the specificity of protein-protein interactions. Negative controls could include non-specific immunoglobulin G (IgG) or unrelated proteins as negative binding controls.

Response: Thank you for pointing out this issue, and we apologize for the inappropriate presentation in images of GST pull-down and immunoprecipitation assays. In the original data, we have used a purified GST protein in GST pull-down assays and IgG in immunoprecipitation to ensure the specificity of the GCLM-NKRF interaction (**response Fig. 9A, B**; Fig. 3b, c in revised manuscript), the NKRF-P65 interaction (**response Fig. 9C**; Fig. 3i in revised manuscript) and the GCLM-importin $\alpha 5$ interaction (**response Fig. 9D**; Supplementary Fig. 4f in revised manuscript). Subsequent relevant experiments were performed using the same experimental protocol, which was described in more detail in the revised “Methods” section (revised

page 30, lines 673-674). We have corrected the presentation in all the images of the GST pull-down and immunoprecipitation assays and hope that this reviewer will agree with our interpretation.

Figure legend 9. (A) GST pull-down analysis detecting the level of purified His-NKRF interacted with purified GST-GCLM or GST. (B) Co-IP analysis showing the level of NKRF bound by GCLM using anti-GCLM or IgG antibody in HCT116 and 293T cells. (C) Control or GCLM knockdown HCT116 and 293T cells overexpressed HA-tagged NKRF. Co-IP analysis showing the interaction of NKRF and p65/p50 using anti-HA or IgG antibody. (D) HCT116 or 293T cells overexpressed Flag-tagged GCLM WT. Co-IP analysis showing the interaction of importin α 5 and GCLM using anti-Flag or IgG antibody.

10. DAPI staining is commonly used to assess nuclear morphology, but it is not a specific marker for cell viability. To accurately measure cell viability, complementary assays such as MTT, MTS, or ATP assays should be performed. These assays directly measure metabolic activity or ATP levels, providing a more reliable indicator of cell viability.

Response: We apologize for the confusion caused by the unclear description of our cell viability assays. In this study, we actually measured cell viability using a CellTiter 96[®] AQueous One Solution Cell Proliferation Assay (MTS) kit (Cat# G3580, Promega) according to the standard instructions. In addition, we also performed complementary ATP assays using a CellTiter-Glo[®] Luminescent Cell Viability Assay kit (Cat# G7570, Promega) to verify the key findings. For example, **response Fig. 10 A-C** shows the same results as our original cell viability assays. These data have been included in revised “Results” section (Supplementary Fig. 3g, 4k, 5i in revised manuscript). We hope that you to concur with our interpretation.

Figure legend 10. (A) ATP level were detected in HCT116 cells depleting GCLM with or without importin α 5 inhibition under oxaliplatin treatment. (B) ATP level were detected in GCLM knockdown HCT116 cells overexpressing vector or rGCLM WT with control or importin α 5 silencing under oxaliplatin treatment. (C) ATP level analysis in GCLM knockdown HCT116 cells overexpressing rGCLM WT or T17A, T17E mutants with or without oxaliplatin treatment.

11. Hematoxylin and eosin (H&E) staining is a standard method for histological analysis, but it may not provide sufficient specificity for identifying fibrosis. Additional staining techniques such as Masson's trichrome or Sirius Red staining, which specifically target collagen fibers, would enable more accurate quantification and characterization of fibrosis in tumor tissues.

Response: Thank you very much for your helpful suggestions. As suggested, we performed Masson's trichrome staining to identify collagen fibers in paraffin-embedded tissue from PDX models². The result revealed that the combination treatment group with GCLM inhibition and oxaliplatin treatment showed more loose and regular collagen fibers (blue staining), than the control group or groups with either GCLM depletion or oxaliplatin treatment alone (**response Fig. 11; Fig 1i and Supplementary Fig. 1j** in revised manuscript). These data have been included in the revised manuscript (revised page 7, lines 150-151).

Figure legend 11. Representative images of Masson's trichrome staining in PDX #1/2-based paraffin-embedded subcutaneous tumor sections. The collagen fibers stain blue with a red background.

References

1. Wang Y, *et al.* Inhibition of fatty acid catabolism augments the efficacy of oxaliplatin-based chemotherapy in gastrointestinal cancers. *Cancer Lett* **473**, 74-89 (2020).
2. He X, *et al.* Extracellular matrix physical properties govern the diffusion of nanoparticles in tumor microenvironment. *Proc Natl Acad Sci U S A* **120**, e2209260120 (2023).

Manuscript ID: NCOMMS-24-17740A

Title: Nucleus-Translocated GCLM Promotes Chemoresistance in Colorectal Cancer through a Moonlighting Function

Detailed point-by-point responses to the reviewers' comments:

Reviewer #2 (Remarks to the Author):

The authors have addressed all of my concerns—congratulations!

Response: We thank the reviewer for providing valuable comments to enhance the quality of our manuscript. We are grateful for your approval of our modifications.

Reviewer #3 (Remarks to the Author):

Authors have successfully responded to all comments of the reviewer.

Response: Thank you very much for your comments and suggestions, which were very constructive and helpful in strengthening our manuscript. We also thank your satisfaction with our revised results.